# Maternal regulation of biliary disease in neonates via gut microbial metabolites

Jai Junbae Jee[1,14,17], Li Yang[1,17], Pranavkumar Shivakumar [1,2,17], Pei-pei Xu[3,17], Reena Mourya[1], Unmesha Thanekar [1,15], Pu Yu[4], Yu Zhu[5], Yongkang Pan[4], Haibin Wang[6], Xufei Duan[6], Yongqin Ye[7], Bin Wang[7], Zhu Jin[8], Yuanmei Liu[8], Zhiqing Cao[9], Miki Watanabe-Chailland[10], Lindsey E. Romick-Rosendale[10], Michael Wagner[3,11], Lin Fei [3,12], Zhenhua Luo[1,16], Nicholas J. Ollberding[12,13], Shao-tao Tang[2✉] & Jorge A. Bezerra [1,2✉]

Maternal seeding of the microbiome in neonates promotes a long-lasting biological footprint, but how it impacts disease susceptibility in early life remains unknown. We hypothesized that feeding butyrate to pregnant mice influences the newborn's susceptibility to biliary atresia, a severe cholangiopathy of neonates. Here, we show that butyrate administration to mothers renders newborn mice resistant to inflammation and injury of bile ducts and improves survival. The prevention of hepatic immune cell activation and survival trait is linked to fecal signatures of Bacteroidetes and Clostridia and increases glutamate/glutamine and hypoxanthine in stool metabolites of newborn mice. In human neonates with biliary atresia, the fecal microbiome signature of these bacteria is under-represented, with suppression of glutamate/glutamine and increased hypoxanthine pathways. The direct administration of butyrate or glutamine to newborn mice attenuates the disease phenotype, but only glutamine renders bile duct epithelial cells resistant to cytotoxicity by natural killer cells. Thus, maternal intake of butyrate influences the fecal microbial population and metabolites in newborn mice and the phenotypic expression of experimental biliary atresia, with glutamine promoting survival of bile duct epithelial cells.

[1] Divisions of Gastroenterology, Hepatology and Nutrition and The Liver Care Center at Cincinnati Children's Hospital Medical Center, Cincinnati, OH 45229, USA. [2] Department of Pediatrics, University of Cincinnati, College of Medicine, Cincinnati, OH 45267, USA. [3] Department of Pediatric Surgery, Union Hospital, Tongji Medical College, Huazhong University of Science and Technology, 430022 Wuhan, Hubei, China. [4] Department of Neonatal Surgery, Xi'an Children's Hospital, 710003 Xi'an, Shaanxi, China. [5] Department of Pediatrics, Western China Second Hospital, Sichuan University, 610041 Chengdu, Sichuan, China. [6] Department of Pediatric Surgery, Wuhan Children's Hospital, Tongji Medical College, Huazhong University of Science and Technology, 430015 Wuhan, Hubei, China. [7] Department of General Surgery, Shenzhen Children's Hospital, 518038 Shenzhen, Guangdong, China. [8] Department of Pediatric General Thoracic and Urology Surgery, The Affiliated Hospital of Zunyi Medical University, 563000 Zunyi, Guizhou, China. [9] Department of Pediatric Surgery, Jiangmen Maternity and Child Health Care Hospital, 529000 Jiangmen, Guangdong, China. [10] Division of Pathology, Cincinnati Children's Hospital Medical Center, Cincinnati, OH 45229, USA. [11] Division of Biomedical Informatics, Cincinnati Children's Hospital Medical Center, Cincinnati, OH 45229, USA. [12] Division of Biostatistics and Epidemiology, Cincinnati Children's Hospital Medical Center, Cincinnati, OH 45229, USA. [13] Department of Rehabilitation, Exercise, and Nutrition Sciences, University of Cincinnati, College of Medicine, Cincinnati, OH 45267, USA. [14] Present address: Department of Pediatrics, Severance Children's Hospital, Yonsei University College of Medicine, Yonsei-ro, Seodaemun-gu, Seoul, Republic of Korea. [15] Present address: Department of Bone Marrow Transplant and Cellular Therapy, St. Jude Children's Research Hospital, Memphis, TN 38105, USA. [16] Present address: Institute of Precision Medicine, The First Affiliated Hospital, Sun Yat-sen University, Guangzhou City, Guangdong, China. [17] These authors contributed equally: Jai Junbae Jee, Li Yang, Pranavkumar Shivakumar, Pei-pei Xu. ✉email: tshaotao83@hust.edu.cn; jorge.bezerra@cchmc.org

The onset of diseases in early postnatal life infers pathogenic mechanisms that are genetically determined or depend on maternal factors that hinder the adaptation of the neonate to the extrauterine environment. A non-genetic factor associated with disease susceptibility is the maternal microbiome, whose changes have been linked to cardiovascular[1,2], allergic[3], and metabolic diseases[4] later in life. However, how the maternal microbiome makes newborns vulnerable to disease early in life remains largely underexplored[5].

The potential role of the gut microbiome as a non-genetic cause of the neonatal disease is inferred from an earlier report that antibiotic treatment of pregnant mice enriched the neonatal intestine with butyrate-producing *Anaerococcus lactolyticus* and rendered newborn mice resistant to biliary atresia (BA). In humans, BA is a disease of multifactorial pathogenesis that manifests biochemically at birth and as a full clinical syndrome within 1–3 months of life. Without curative treatment, the disease progresses to cirrhosis and represents the most common cause of end-stage liver disease in childhood[6]. Like in humans, the susceptibility of mice to virus-induced BA is restricted to the immediate postnatal period and exhibits an epithelial injury dependent on Th1 and Th17 cells and soluble factors, followed by a predominant Th2-driven reparative response[7]. The early onset of disease and the lack of genetic defects in most patients raise the possibility that maternal and other prenatal factors influence the susceptibility of disease. Based on a previous report that antibiotic treatment of pregnant mothers enriched the microbiome of offspring with butyrate-producing bacteria and rendered newborn mice resistant to biliary injury[6], we tested the hypothesis that mothers regulate the susceptibility of biliary disease in neonates by influencing the composition and function of the intestinal microbiome. Here we report that the enrichment of the intestinal microbiome of newborn mice from butyrate-fed dams with Bacteroidetes and Clostridia and an increase of fecal glutamate/ glutamine render neonates resistant to experimental BA. This microbiome and metabolite signature is under-represented in human infants with BA. The protection against biliary injury is assigned to glutamine by making bile duct epithelial cells resistant to natural killer (NK) cell-mediated cytotoxicity.

## Results

**Maternal intake of butyrate protects newborn offspring from bile duct injury.** We fed butyrate to pregnant mice and evaluated the outcome of newborn mice to virus infection. Butyrate was added to drinking water (200 mmol/L, pH adjusted to 7.5) fed to female BALB/c mice throughout pregnancy and the first 4 weeks after delivery; a separate group received only drinking water and served as controls (Fig. 1A). Within 24 h of delivery, we administered $1.5 \times 10^6$ fluorescence-forming units (ffu) of *Rhesus* rotavirus (RRV) intraperitoneally to the offspring (or similar volume as phosphate-buffered saline [PBS] to control newborn mice). In mice from water-fed mothers, rotavirus infection resulted in progressive jaundice and 80% mortality within 14 days, with the remaining 20% representing the small variability in severity that can be observed in this mouse model. In contrast, the clinical evolution of mice from butyrate-fed mother segregated into a small group of mice with a similar course of progressive jaundice and mortality (40% of pups, referred to as "diseased") and a second survival group that showed only transient jaundice (60% of pups, referred to as "resistant"). Pups in this "resistant" group had lower levels of serum total bilirubin (a marker of cholestasis) and alanine aminotransferase (ALT; a marker of liver injury) than symptomatic littermates and controls (Fig. 1B–E). In keeping with the clinical and biochemical improvement, extrahepatic bile ducts (EHBD) of pups from

butyrate-fed mothers were patent and the liver inflammation was restricted to portal tracts, while rotavirus-infected controls had complete duct obstruction and extensive liver injury (Fig. 1F and Fig. S1A–C). The decreased tissue inflammation and injury in mice from butyrate-fed mothers was independent of viral titers in liver or bile ducts at day 7 after RRV infection (Fig. 1G).

To control for a potential influence of butyrate-induced acidification of the intestinal contents on the protection against the disease phenotype, we fed propionic acid (sodium propionate, 200 mmol/L in drinking water, pH adjusted to 7.5) to pregnant mice according to the same experimental protocol used for butyrate. We found that propionic acid feeding during gestation did not protect neonatal mice from RRV-induced BA, as evidenced by the uniform onset and progression of jaundice, 100% mortality by day 14, high serum ALT and bilirubin, and complete obstruction of EHBD and portal inflammation (Fig. S2A–F). These experiments supported a likely causative effect of maternal butyrate feeding in the protection of newborn mice to biliary injury.

Based on the immunomodulatory properties of butyrate, we quantified hepatic mononuclear cells (MNCs) 7 days after rotavirus infection (time of duct obstruction) by flow cytometric analyses and found significant decreases in helper T cells ($T_H$, CD3$^+$CD4$^+$), cytotoxic T ($T_C$, CD3$^+$CD8$^+$), and NK (CD3$^-$CD49b$^+$) cells, with preferential increases in regulatory T cells ($T_{REG}$, CD3$^+$CD4$^+$CD25$^+$ Foxp3$^+$) and IL-10$^+$ $T_{REG}$ cells, but no effect on macrophages, neutrophils, or dendritic cells (DCs; Fig. 1H and Fig. S3). The gating strategy for flow cytometric analyses is shown in Fig. S10A–G. Thus, the maternal intake of butyrate suppressed the tissue infiltration of effector lymphocytes in neonates infected with RRV, promoted the activation of IL-10$^+$ $T_{REG}$ cells, and prevented the expression of the disease phenotype.

To determine whether butyrate directly promoted disease resistance and suppression of the inflammatory response to RRV, we administered butyrate (0.3 mg/g of body weight) orally to RRV infected newborn mice beginning 1 day after infection (Fig. 2A) following a previously published protocol[8]. Butyrate lowered the incidence of jaundice, increased survival rate, and prevented duct obstruction without altering viral titers in bile ducts and livers (Fig. 2B–E). Based on these findings and on the immunomodulatory properties of butyrate[9], we tested the hypothesis that butyrate suppresses the activation of effector lymphocytes by culturing hepatic MNCs from RRV-infected neonatal mice in the presence of 0.1–0.25 mM butyrate for 3 days[10–12]. Butyrate had only a modest effect on *Foxp3* mRNA expression in MNCs and unexpectedly decreased *Il10* mRNA (Fig. 2F, G). These findings raised the possibility that the protective effects of maternal feeding of butyrate against hepatobiliary injury included other microbiome-dependent factors.

**Disease-resistant offspring shares a similar microbial signature with butyrate-fed mothers.** To investigate whether butyrate produced a selective enrichment of the intestinal microbiota linked to the suppression of tissue injury, we analyzed the stool microbiome from mothers at the time of delivery and from RRV-infected and -naive neonatal mice 12–14 days after virus inoculation (Fig. 3A). In the water-control group, non-metric multidimensional scaling (NMDS) analysis of operational taxonomic units (OTUs) generated by 16s rRNA sequencing showed that the microbiome of the mother segregated with that of RRV-naive offspring and differed from infected newborn mice with biliary obstruction (Fig. 3B, C). Similarly, the microbial OTUs from the butyrate group that did not develop biliary obstruction after RRV ("resistant" subgroup) were similar to their butyrate-fed mothers

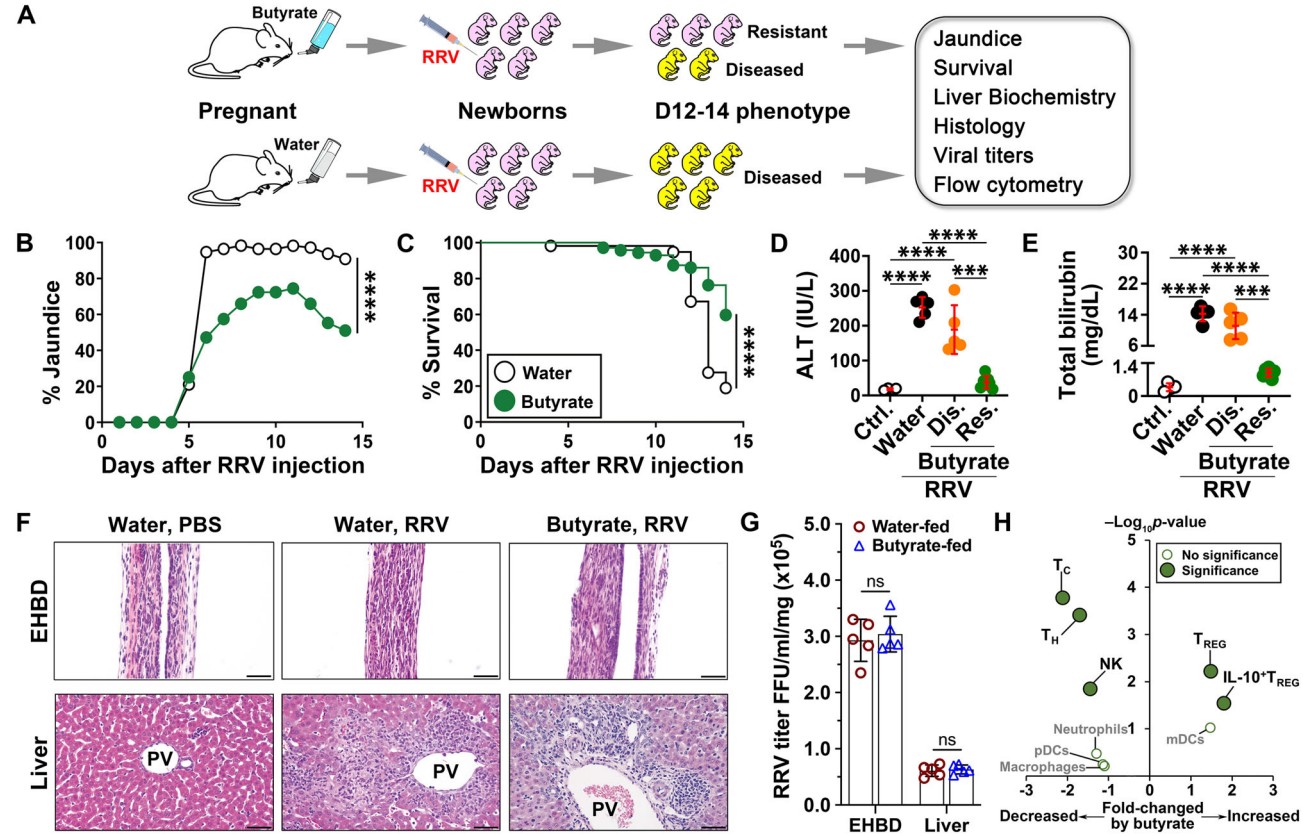

**Fig. 1 Maternal intake of butyrate suppresses liver and bile duct injury in the offspring. A** Schematic representation showing butyrate or water administration followed by evaluation of biliary disease in rotavirus (RRV)-infected mice. **B** Jaundice (generalized linear mixed effect model with logit link and two-sided Wald test with Bonferroni correction; ****$p < 0.0001$) and **C** survival (two-sided log-rank test; ****$p < 0.0001$) rates in RRV-infected newborn mice from water- or butyrate-fed mothers ($n = 67$–74 mice per group). Plasma alanine aminotransferase (ALT, [**D**]) and total bilirubin (**E**) from newborn mice 12–14 days after RRV ($n = 5$) or phosphate-buffered saline (PBS; Ctrl, $n = 4$) from water- or butyrate-fed mothers (bile duct injury/obstruction = Dis., $n = 5$; asymptomatic/resistant = Res., $n = 7$; mean ± SD, two-tailed ANOVA with Duncan's multiple comparison; ***$p < 0.001$, ****$p < 0.0001$). **F** Extrahepatic bile duct (EHBD) and liver sections 12–14 days after RRV or PBS (water and butyrate = maternal feeding; magnification bar = 100 μm; PV portal vein). In all, 15–30 EHBD and 5–10 liver sections (corresponding to >100 sections at ×200 or ×400 magnification fields from $n = 11$–22 mice) stained with H&E per tissue specimen were evaluated for histology analysis. **G** Virus titers in EHBD and livers at day 7 after RRV infection of newborn mice from water and butyrate-fed dams (mean ± SD, two-tailed unpaired Student's $t$ test with Welch's correction; $n = 5$ biologically independent EHBD or livers per group; ns = not significant). **H** Volcano plot illustrating hepatic mononuclear cells with $p$ values and fold changes between neonatal mice from water- and butyrate-treated mothers 7 days after RRV infection. The replicate values were determined using biologically distinct samples and $p$ values calculated using unpaired Student's $t$ test with two-tailed distribution. The gating strategy for flow cytometric analyses is shown in Fig. S10A–K. Source data for this figure are provided as a Source data file.

and to RRV-naive newborn mice but differed from the subgroup of mice that developed the disease ("diseased subgroup"; Fig. 3D, E). Additional evidence of a shared bacterial signature between butyrate-treated pregnant mice and their disease-resistant offspring included a higher number of shared OTUs (Fig. 3F and Table S1), a predominant enrichment of Firmicutes and Bacteroidetes, and a decrease in Proteobacteria in mice resistant to RRV-induced biliary injury when compared to those with the diseased phenotype (Fig. 3G, H). Correspondingly, analysis of the 1315 OTUs detected exclusively in fecal specimens from butyrate-fed pregnant mice showed a predominance of Bacteroidetes and Firmicutes (Table S2). These data linked the microbiome of butyrate-fed mothers with their offspring uniquely resistant to the disease phenotype.

**Murine fecal metabolites suppress immune cell activation.** In search of potential mechanisms used by the microbiome to suppress the disease phenotype, we filtered fecal supernatants from neonatal mice from water- or butyrate-fed mothers and cultured the supernatants with hepatic MNCs from RRV-infected neonatal mice (Fig. 4A). Fecal supernatants from the butyrate group significantly increased *Il10* and *Foxp3* mRNAs and decreased *Tnfa* expression in MNCs (Fig. 4B–D), suggesting that stool-derived molecules suppressed pro-inflammatory circuits. To identify these molecules, we analyzed the fecal supernatants from neonatal mice from water- or butyrate-fed mothers by proton nuclear magnetic resonance ($^1$H-NMR) spectroscopy-based metabolomics customized to quantify 56 compounds. NMDS analysis of the metabolites distinguished butyrate from non-butyrate groups (Fig. 4E), with a predominant increase in amino acids, ketone bodies, short-chain fatty acids, and metabolic intermediates of the citric acid cycle in fecal supernatants of pups born to butyrate-fed dams (Fig. 4F and Fig. S4A, B). Of the 37 metabolites that differed significantly between the two groups, the abundance of hypoxanthine and glutamate showed the largest increase in fecal supernatants from the butyrate group, with their related molecules inosine and glutamine changing at a lower magnitude (Fig. 4G), thus raising the potential for their effector role in the suppression of the immune-mediated injury of bile

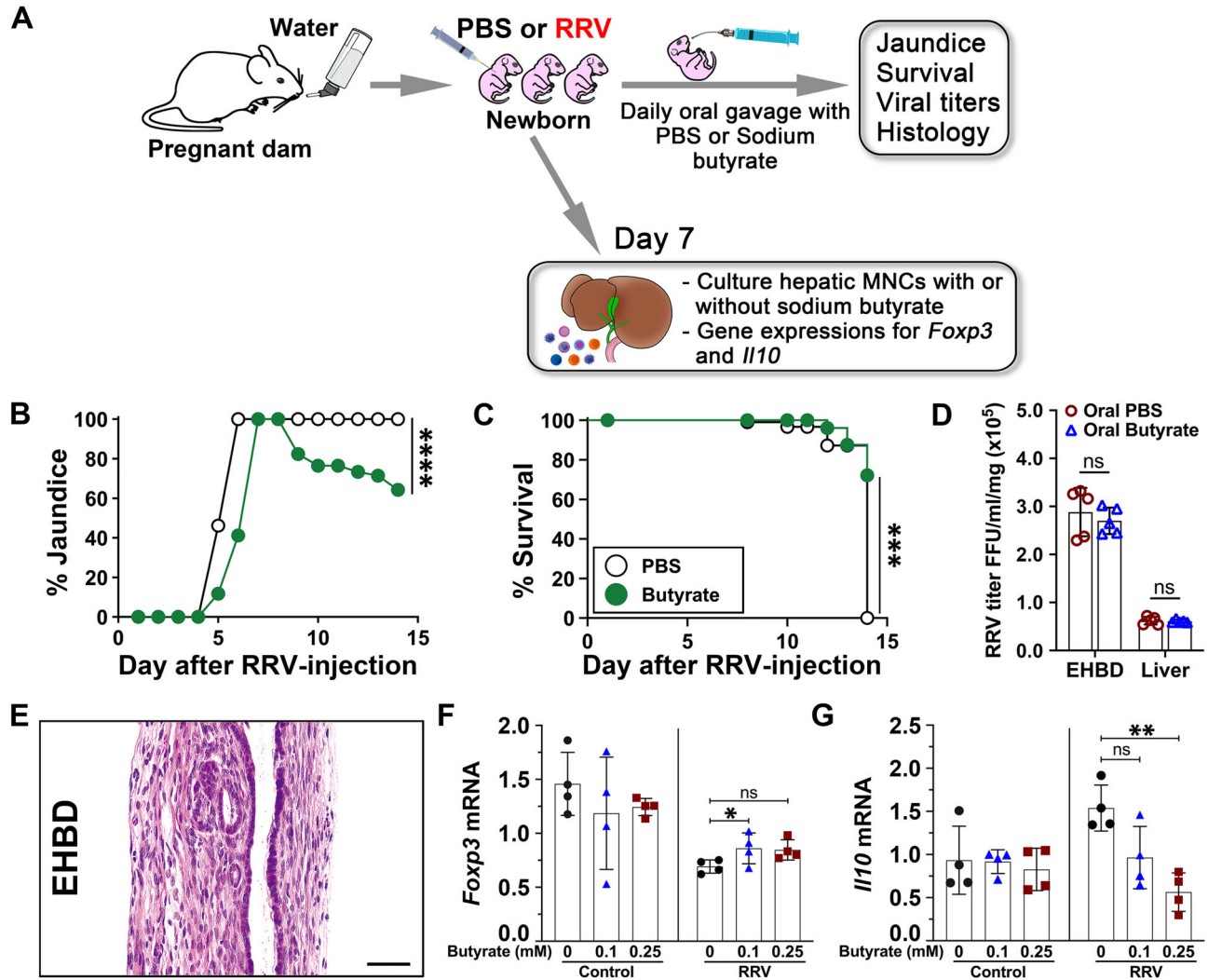

**Fig. 2 Treatment of neonates with butyrate decreases hepatobiliary injury. A** Diagrammatic outline of gavaging RRV-infected neonatal mice with sodium butyrate. **B** Jaundice (generalized linear mixed effect model with logit link and two-sided Wald test with Bonferroni correction; ****$p < 0.0001$) and **C** survival (two-sided log-rank test; ***$p < 0.001$) rates in RRV-infected newborn mice treated daily with butyrate or PBS. **D** Virus titers in EHBD and livers at day 7 after RRV infection of newborn mice from water-fed dams (mean ± SD, two-tailed unpaired $t$ test with Welch's correction; $n = 5$ per group; ns = not significant) and **E** section of EHBD from butyrate-treated mice 14 days after RRV. In all, 15–30 EHBD sections (corresponding to >100 sections at ×200 or ×400 magnification fields from $n = 11$ mice) stained with H&E per tissue specimen were evaluated for histology analysis. Scale bar = 50 µM. *Foxp3* (**F**) and *Il10* (**G**) mRNA in RRV-naive or primed hepatic mononuclear cells cultured with or without butyrate, normalized to *Gapdh* (mean ± SD, two-tailed ANOVA with Duncan's multiple comparisons, $n = 3$ per group. *$p < 0.05$, **$p < 0.01$, ns = not significant). Source data for this figure are provided as a Source data file.

ducts in neonatal mice. To examine the relationship between the metabolites and the fecal microbial population, we analyzed the 16s rRNA-sequencing data of fecal microbiome from "diseased" and "resistant" subgroups using PICRUSt to quantify host gene families, MetaCyc to mine pathways, and ALDEx2 to test for differential abundance. We found an enrichment of glutamate/glutamine, hypoxanthine/inosine, and butyrate pathways in the "resistant" subgroup of mice (Fig. S5). The butyrate pathways were represented by butyrate-producing Firmicutes (Firmicutes, Clostridia, Clostridiales and *Ruminococcaceae*) and Bacteroidetes (Bacteroidetes, Bacteroidia, Bacteroidales, *Bacteroidaceae*, *Bacteroides*) taxa (Fig. S6). The bacterial genes common to these pathways were linked to molecules of the Acetyl-CoA and Lysine pathways of butyrate synthesis (Table S3).

**Metabolic pathway enrichment in fecal metagenomic sequencing of human newborns**. To explore the relevance of the microbial and metabolic signatures of neonatal mice to potential

mechanisms of hepatobiliary injury in human infants, we performed shotgun metagenomic sequencing of stool samples collected from 102 infants at the time of diagnosis of BA (median age in days [interquartile range (IQR)]: 48, [35, 61]) and from 28 normal controls (NC, age: 64 [33, 79]) (Fig. 5A and Fig. S7A). There were no differences between BA and control groups in the number of observed species or Shannon diversity or segregation based on the overall microbial community composition (Fig. S7B–E). This lack of difference in Shannon index and NMDS may be related to the inherently low diversity of intestinal microbiome in young children[13]. Examining the bacterial population by regression analysis identified a greater abundance of Proteobacteria, Bacilli (*Lactobacillus* spp.), Fusobacteria, and other pathogens such as *Streptococcus* spp., *Klebsiella* spp., and *Enterococcus* spp. in BA, while NCs had significant enrichment of Bacteroidetes and Clostridia (Fig. 5B and Fig. S7F, G) akin to the enrichment observed in disease-resistant newborn mice from butyrate-fed mothers. To determine how this microbial signature

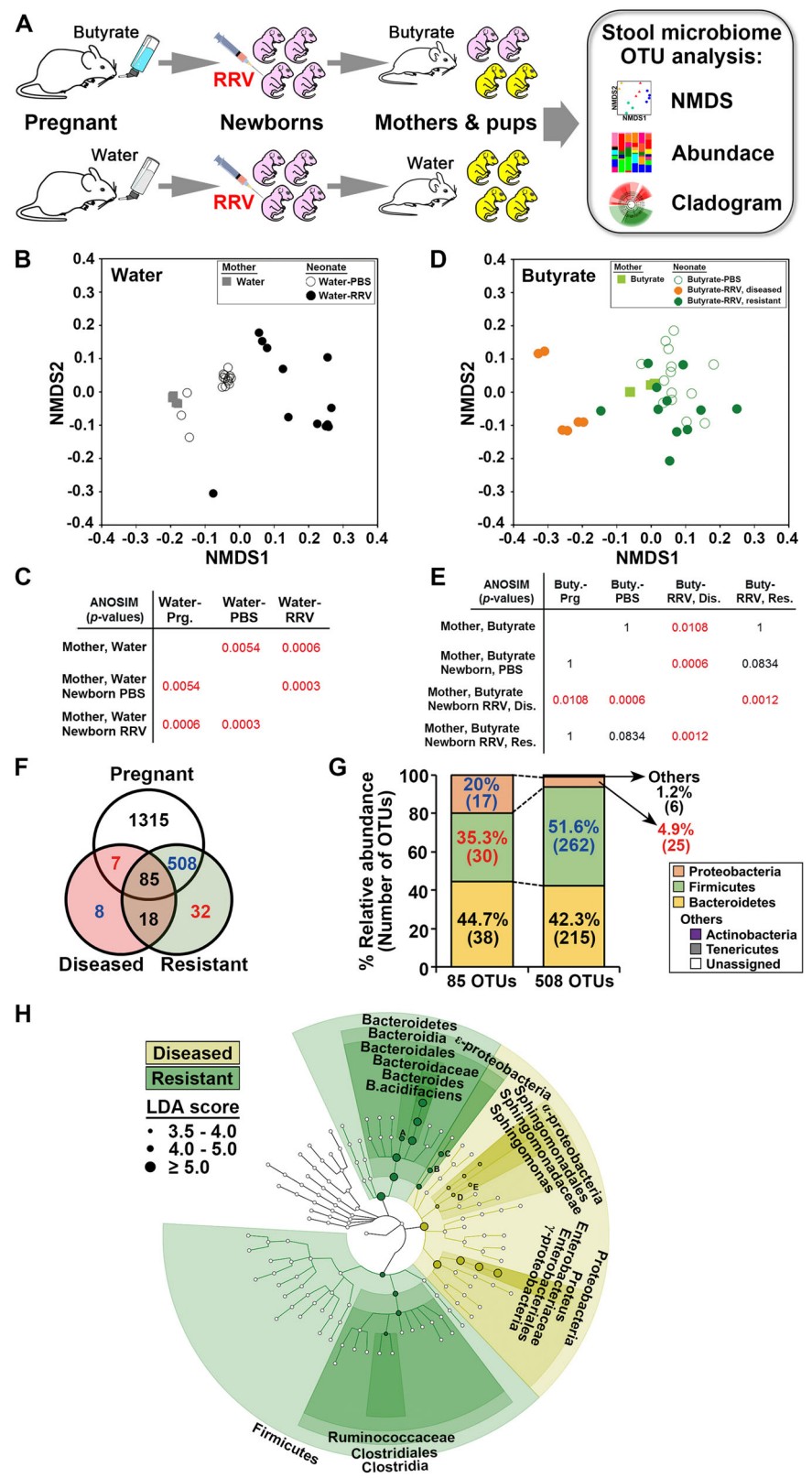

potentially translates to differences in function, we aggregated sequence reads of the human microbial mapping of UniRef90 gene families to MetaCyc pathways using HUMAnN2 and tested for differential abundance using ALDEx2. Among the 14 pathways identified based on adjusted $p$ values <0.05, seven were associated with glutamate/glutamine and butyrate molecules, with the highest effect size for PWY-4984 (urea cycle) and

CITRULBIO-PWY (L-citrulline biosynthesis) segregating with NCs (Fig. 5C and Table S4). Of the seven species enriched in NCs (Fig. S7G), *Flavonifractor plautii*, *Hungatella hathewayi*, *Clostridium neonatale*, *Bacteroides dorei*, and *Bacteroides fragilis* (Fig. S7H) showed overexpression of enzymes and genes involved in butyrate production when analyzed using BioCyc databases (Table S5). In contrast to the high concentration of hypoxanthine

**Fig. 3 Shared microbial signatures between butyrate-fed mother and offspring resistant to the disease phenotype. A** An experimental overview of stool microbiome analysis from RRV-infected mice from water- and butyrate-fed females. Non-metric multidimensional scaling (NMDS) ordinations and analysis of similarities (ANOSIM) of 16s rRNA organismal taxonomic units (OTUs) of fecal specimens from water- (**B**, **C**) and butyrate-fed (**D**, **E**) mothers and offspring 12–14 days after RRV or PBS injection. **F** Venn diagram depicting the number of OTUs from butyrate-treated pregnant female and their offspring infected with RRV with (diseased) or without (resistant) biliary obstruction (Fisher's exact test and Z-score value from adjusted standardized residuals, $n = 6$–11 per group). **G** Bacterial communities in 85 OTUs shared between butyrate-fed mothers and diseased and resistant offspring compared to 508 OTUs shared between butyrate-fed mothers and resistant offspring (Chi-square test and Z-score value from adjusted standardized residuals, $n = 6$–11 per group). **H** Cladogram showing a significantly different abundance of bacterial taxa with LDA scores >3.5 magnitude changes between diseased and resistant phenotypes in RRV-infected newborn mice from butyrate-fed mothers (Kruskal–Wallis sum-rank test, $n = 6$–11 per group). Source data for this figure are provided as a Source data file.

in stools of RRV-naive mice from butyrate-fed mothers, hypoxanthine (in the PWY0-1297 super-pathway of purine degradation) was enriched in stools of infants with BA, thus raising the possibility that glutamate/glutamine and hypoxanthine have different roles in disease phenotype determination.

**Glutamine prevents biliary injury and promotes the survival of epithelial cells.** To directly investigate the biological roles of glutamate/glutamine[14–16] and hypoxanthine/inosine[17] in mechanisms of disease, we administered glutamine (0.25 mg/g of body weight), inosine (a nucleoside composed of ribose ring and hypoxanthine, 0.2 mg/g of body weight), glutamine + inosine, or PBS intraperitoneally every other day to newborn mice from water-fed mothers beginning one day after RRV inoculation (Fig. 6A). The rationale for selecting glutamine was based on its low levels in children with BA[18], its role as a conditionally essential amino acid in influencing intracellular bio-energy utilization[19], and its support for intracellular redox homeostasis via glutathione (GSH)[20,21]; in contrast, glutamate (with its negative charge), exerts extrahepatic biological effects such as in neurotransmission[22]. Due to the poor water solubility and potential toxicity of hypoxanthine, we used inosine, a naturally occurring purine and precursor of hypoxanthine with a better safety profile and immunoregulatory properties[23].

Administration of inosine did not change the development of jaundice or the poor survival of neonatal mice infected by RRV; in contrast, glutamine alone significantly decreased jaundice and increased survival to 83.3% (Fig. 6B–G) and suppressed tissue injury (Fig. 6H–O), with a low population of T, NK cells, DCs, neutrophils, and macrophages and an increase in IL-10+ T_REG cells (Fig. 7A, B and Fig. S8). Functional assessment of hepatic NK cells isolated from RRV-infected mice treated with glutamine showed a decrease in their ability to lyse cholangiocytes in vitro (bile duct epithelial cells; Fig. 7C).

To investigate how glutamine may regulate the ability of NK cells to lyse cholangiocytes, we used a NK cell–cholangiocyte co-culture assay to first assess whether interleukin (IL)-10 protects against cell lysis. The addition of IL-10 (30 ng/ml) to the culture media of RRV-primed hepatic NK cells and cholangiocytes did not prevent cholangiocyte lysis (Fig. 7D). To investigate the mechanisms by which glutamine suppresses cell lysis by NK cells, we performed the assay with glutamine and found that 8–16 mM glutamine blocked cholangiocyte lysis (Fig. 7E). To precisely identify the cellular target for glutamine, we incubated RRV-primed hepatic NK cells or cholangiocytes with glutamine separately prior to the cell lysis assay. While preincubation of NK cells with glutamine did not suppress cell lysis (Fig. 7F), the preincubation of cholangiocytes with glutamine rendered cells resistant to NK cell-induced lysis (Fig. 7F).

Based on the ability of glutamate/glutamine to modulate intracellular oxidative balance, we measured GSH in cholangiocytes cultured with glutamine and found a significant increase in GSH at all concentrations (Fig. 7G). To examine whether the enrichment of GSH would independently prevent cholangiocyte

lysis, we added N-acetyl cysteine (NAC) to the co-culture assay and observed near-complete prevention of cholangiocyte lysis (Fig. 7H). To test the relevance of GSH abundance to the protective effect of glutamine in vivo, we inoculated RRV into new groups of neonatal mice and assigned them to glutamine or saline treatment using the experimental strategy and protocol depicted in Fig. 6A. While RRV infection significantly decreased the concentration of GSH in the liver and EHBD, the administration of glutamine increased GSH levels in both tissues and approached the levels of uninfected controls (Fig. 7I, J). Altogether, these data identified a uniquely protective role of glutamine in promoting the survival of cholangiocytes against virus-activated NK cells via GSH enrichment.

**Discussion**
The beneficial impact of butyrate feeding during gestation on the suppression of bile duct injury of newborn offspring underscores the relevance of maternal factors in disease susceptibility in early postnatal life. In experimental BA, a prototype of severe disease of prenatal onset and multifactorial pathogenesis, the simple addition of a short-chain fatty acid (butyrate) to the drinking water of pregnant mice rendered newborn mice resistant to tissue injury and the obstructive phenotype of bile ducts. Searching for the biological processes responsible for the improved phenotype, we found similar signatures of the fecal microbiome in butyrate-fed mothers and newborns resistant to experimental BA, as well as an enrichment of glutamate/glutamine and hypoxanthine/inosine metabolites in the intestinal contents of newborns from butyrate-fed mothers. The administration of glutamine to RRV-infected pups and its addition to cultured cholangiocytes resulted in the suppression of inflammation, increased tissue levels of GSH, and protection of epithelial cells against the cytolytic properties of activated hepatic NK cells.

The administration of butyrate directly to neonates shows its ability to suppress the activation of lymphocytes[24], increase IL-10+ T_REG cells[9], and prevent biliary injury independent of microbiome changes. However, its inability to suppress activated hepatic MNCs or induce the expression of Il10 mRNA in vitro pointed to the existence of other factors preventing injury to the liver and bile duct. In keeping with the demonstrated influence of butyrate in promoting differential growth of microbial population in an anaerobic environment[25], stools of newborn mice resistant to tissue injury were populated by Firmicutes and Bacteroidetes (similarly to the fecal microbial signature of their butyrate-fed mothers), a signature that was also present in stools of healthy human infants. Of note, approximately 85% of the total butyrate-producing capacity in the colon is represented by Firmicutes (including *Lachnospiraceae* and *Ruminococcaceae*) and to a lesser extent by the Bacteroidetes lineage[26,27]. An initial cue to the protective role of this microbial signature emerged from its under-representation in the stools of RRV-infected newborn mice with BA and in human infants with the disease, which had a more prominent enrichment of Proteobacteria, Bacilli (*Lactobacillus*

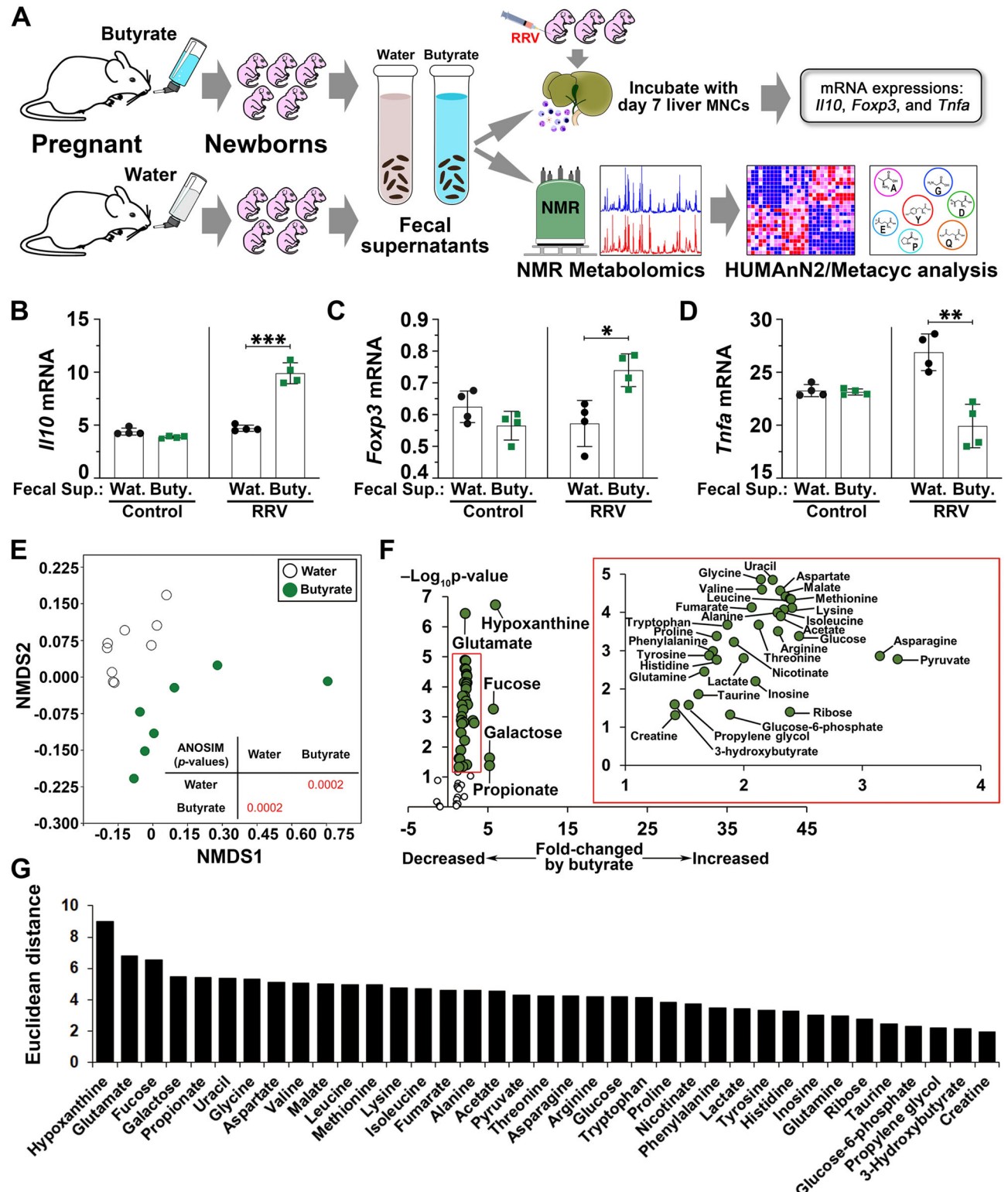

**Fig. 4 Murine fecal metabolites suppress activated immune cells and are enriched with hypoxanthine/inosine and glutamate/glutamine. A** Schematic illustration of in vitro fecal supernatant–immune cell cultures and stool metabolite analysis. mRNA expression for *Il10* (**B**), *Foxp3* (**C**), and *Tnfa* (**D**) as a ratio to *Gapdh* in RRV-primed hepatic MNCs cultured in the presence of fecal supernatants from neonatal mice of water- or butyrate-fed mothers (mean ± SD, two-tailed unpaired Student's *t* test, *n* = 4 per group; *$p < 0.05$, **$p < 0.01$, ***$p < 0.001$). **E** NMDS and ANOSIM of metabolites in fecal supernatants of neonatal mice from water- or butyrate-fed mothers at 14 days of age. **F** Volcano plot illustrating fecal metabolites with *p* values and fold changes between neonatal mice from water- and butyrate-fed mothers (inset depicts metabolites of lower distance). The replicate values were determined using biologically distinct samples and *p* values calculated using unpaired Student's *t* test with two-tailed distribution. **G** Fecal metabolites ordered by Euclidean distance measured from the volcano plot. Source data for this figure are provided as a Source data file.

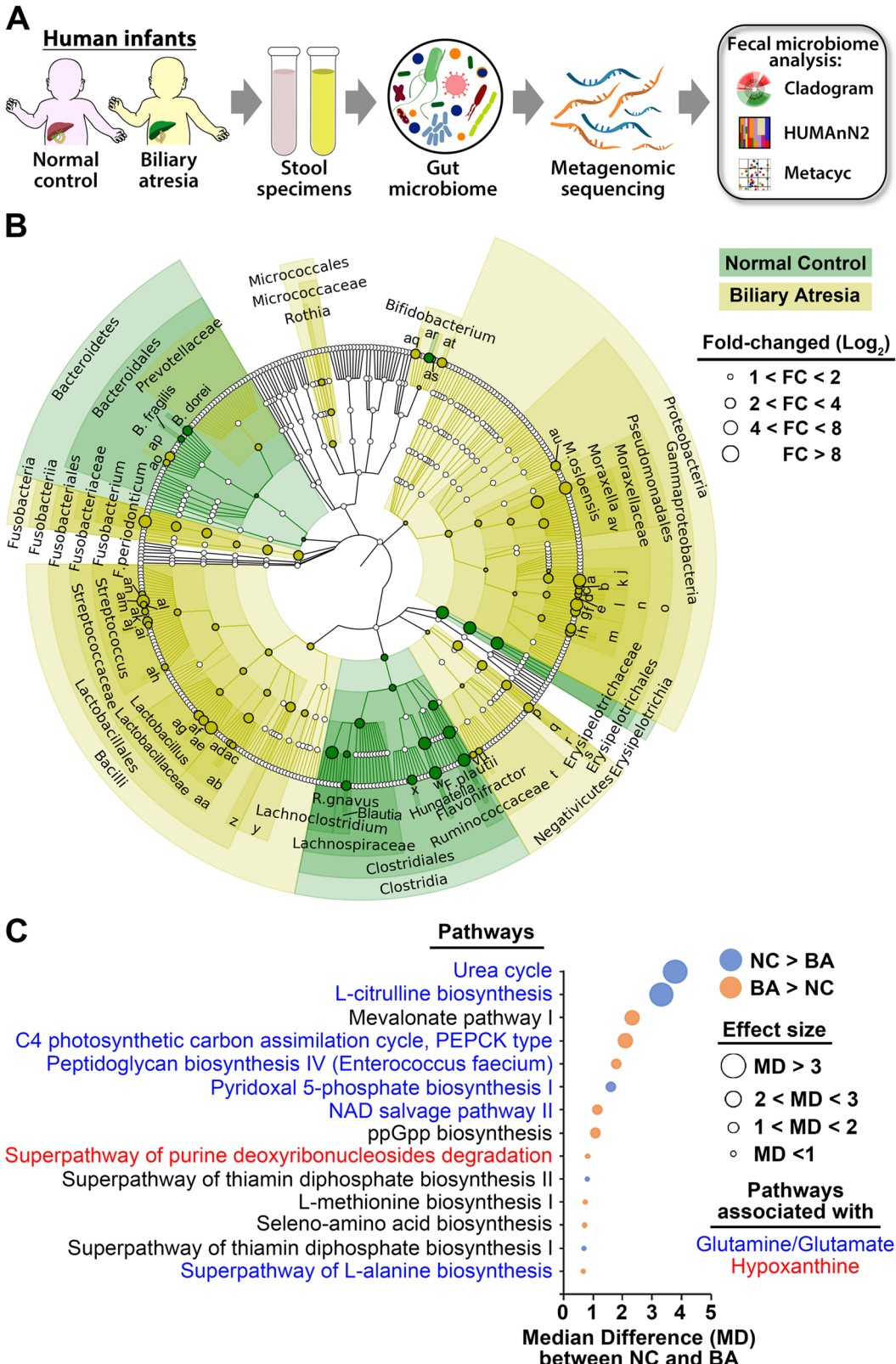

**Fig. 5 Microbiome analysis and functional pathways in human fecal specimens. A** Schematic overview of fecal microbiome analysis from infants with biliary atresia and healthy controls. **B** Cladogram showing a significantly different abundance of bacterial taxa with Log₂ scaled fold changes between infants with biliary atresia ($n = 102$) and age-matched controls ($n = 28$) (adjusted $p$ value <0.05). **C** Functional pathways significantly regulated by microbial community of biliary atresia and controls, calculated by HUMAnN2 on the basis of MetaCyC pathways (adjusted $p$ value <0.05). Source data for this figure are provided as a Source data file.

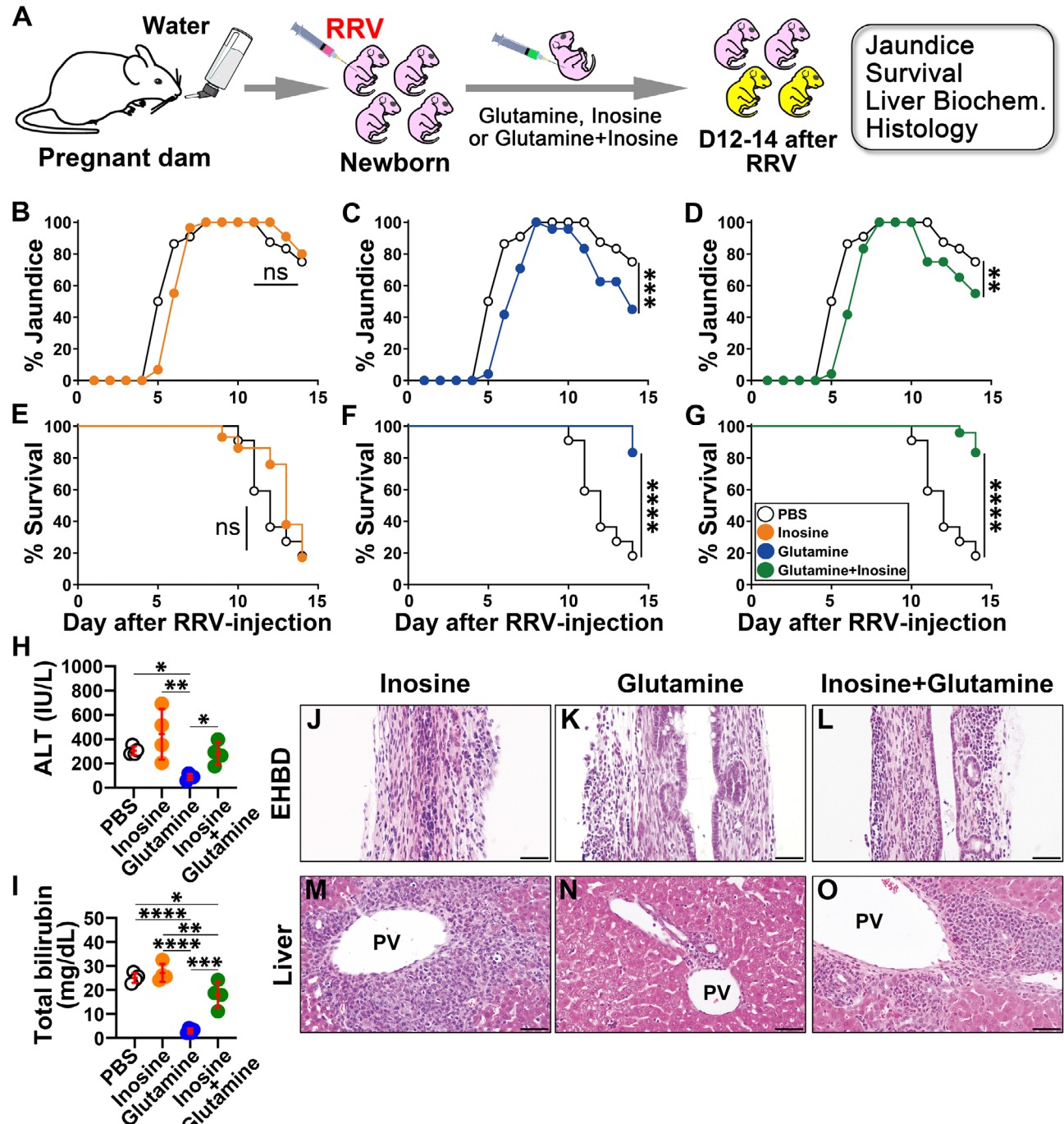

**Fig. 6 Glutamine suppresses the obstructive phenotype of experimental biliary atresia and improves survival. A** An experimental design for the treatment of RRV-infected neonatal mice with inosine and/or glutamine. **B–D** Incidence of jaundice in phosphate-buffered saline (PBS)-injected or RRV-infected newborn mice treated with intraperitoneal doses of inosine, glutamine, and inosine + glutamine ($n = 22$–$29$ per group; generalized linear mixed effect model with logit link and two-sided Wald test with Bonferroni correction. **$p < 0.01$, ***$p < 0.001$, ns = not significant). **E–G** Survival rates of neonatal mice treated with inosine (**E**), glutamine (**F**), and inosine + glutamine (**G**), compared to those receiving PBS (control) during the first 2 weeks of life ($n = 22$–$29$ per group; two-sided log-rank test for survival rate, ****$p < 0.0001$, ns = not significant). Plasma alanine aminotransferase (ALT, [**H**]) and total bilirubin levels (**I**) at days 12–14 after phosphate-buffered saline (PBS; Ctrl, $n = 4$) injection or RRV infection and daily treatment of inosine ($n = 4$), glutamine ($n = 4$), and inosine ± glutamine ($n = 4$; mean ± SD, two-tailed ANOVA with Duncan's multiple comparison; *$p < 0.05$, **$p < 0.01$, ***$p < 0.001$, ****$p < 0.0001$). Hematoxylin–eosin-stained sections of extrahepatic bile ducts (EHBD) and livers sections (**J–O**) from newborn mice 12–14 days after RRV and daily treatment of inosine, glutamine, and inosine ± glutamine 12–14 days after RRV injection (magnification bar = 100 μm; PV portal vein; magnification bar = 100 μm). In all, 15–30 EHBD and 5–10 liver sections (corresponding to >100 sections at ×200 or ×400 magnification fields from $n = 9$–$11$ mice) stained with H&E per tissue specimen were evaluated for histology analysis. Source data for this figure are provided as a Source data file.

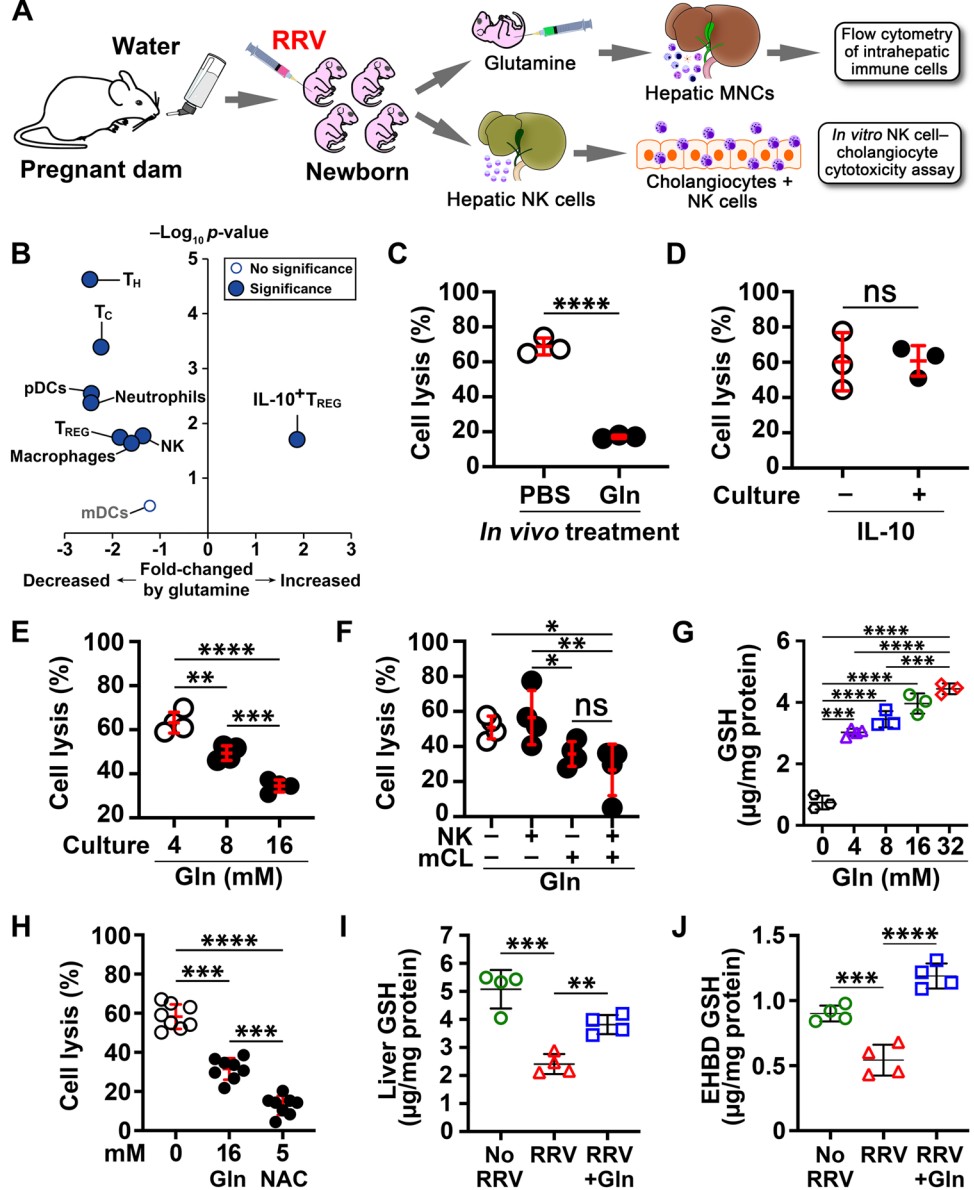

**Fig. 7 Glutamine suppresses hepatic immune cells and promotes cholangiocyte resistance to NK cell-mediated lysis. A** Diagrammatic representation of hepatic mononuclear cell flow cytometry and in vitro NK cell-mediated cholangiotoxicity assay. **B** Volcano plot illustrating hepatic immune cells with $p$ values and fold changes between phosphate-buffered saline (PBS)- and glutamine-injected neonatal mice 7 days after RRV injection. The replicate values were determined using biologically distinct samples and $p$ values calculated using unpaired Student's $t$ test with two-tailed distribution. The gating strategy for flow cytometric analyses is shown in Fig. S10A–K. **C** Percent lysis of cholangiocytes after co-culture with RRV-primed hepatic NK cells from PBS ($n = 3$) or glutamine ($n = 3$) injected neonatal mice and a murine cholangiocyte cell line or **D** in the presence ($n = 3$) or absence ($n = 3$) of 30 ng/ml of IL-10 (mean ± SD, unpaired Student's $t$ test with two-tailed distribution; ****$p < 0.0001$, ns = not significant). Assay of cholangiocyte lysis by RRV-primed NK cells co-cultured with increasing concentrations of glutamine ($n = 4$ per group) (**E**), or **F** when individual cell type is preincubated with glutamine ($n = 4$ per group; mean ± SD, two-tailed ANOVA with Duncan's multiple comparison; *$p < 0.05$, **$p < 0.01$, ***$p < 0.001$, ****$p < 0.0001$, ns = not significant). **G** shows total GSH levels measured in cholangiocytes after incubation with various concentrations of glutamine ($n = 3$–4 per group; mean ± SD, two-tailed ANOVA with Duncan's multiple comparison; ***$p < 0.001$, ****$p < 0.0001$). **H** NK cell-mediated cholangiocyte lysis in the presence of glutamine ($n = 8$; 2 technical replicates and 4 biological replicates) and $N$-acetyl cysteine (NAC) ($n = 8$, 2 technical replicates and 4 biological replicates; mean ± SD, two-tailed ANOVA with Duncan's multiple comparison; ***$p < 0.001$, ****$p < 0.0001$). **I** Total GSH levels measured in supernatants of livers ($n = 4$ biological replicates per group) and **J** extrahepatic bile ducts ($n = 4$ biological replicates per group) from control and RRV-infected neonatal mice with or without glutamine treatment (mean ± SD, two-tailed ANOVA with Duncan's multiple comparison; **$p < 0.01$, ***$p < 0.001$, ****$p < 0.0001$). Source data for this figure are provided as a Source data file.

spp.), Fusobacteria, and other opportunistic pathogens such as *Streptococcus* spp., *Klebsiella* spp., and *Enterococcus* spp[26,28,29]. A stronger cue to the relevance of the microbial population was the increase of glutamate and glutamine in fecal metabolites of neonatal mice born from butyrate-fed mothers and the findings

that glutamine administration largely prevented the development of experimental BA in neonatal mice.

Mechanistically, glutamine displayed a predominant cytoprotective effect of the biliary epithelium against the cytolytic properties of activated hepatic NK lymphocytes. This is consistent

with a previously reported role of NK cells in the injury of the bile duct epithelium, with an antibody-mediated loss of NK cells preventing neonatal mice from developing epithelial injury and obstruction of bile ducts after RRV infection[30]. Here, however, glutamine seems to benefit primarily the bile duct epithelium, as suggested by the preincubation experiments in which the culture of cholangiocytes with glutamine rendered these cells resistant to the cytolytic properties of activated hepatic NK cells. Pursuing potential mechanisms used by glutamine to promote the survival of cholangiocytes, we found that glutamine enriches the intracellular content of GSH in the liver and bile ducts of RRV-infected pups, in cholangiocytes cultured in glutamine-supplemented media, and in promoting the cytoprotection of cholangiocytes cultured in the presence of NAC. The protective role of NAC is in keeping with a previous study reporting a decreased expression of GSH-related genes in children with BA and poor outcome and with the improved outcome of neonatal mice treated with NAC in a model of fibrosing cholangiopathy[31]. Combined, these data point to the important properties of glutamine and GSH in promoting epithelial cell survival.

The directional link of maternal butyrate → enriched neonatal gut Clostridiales and Bacteroidetes and glutamate/glutamine synthesis → protection against tissue injury has implications for understanding pathogenic mechanisms of disease that manifest in early life. Our studies did not control for the influence of cholestasis in the microbiome composition of neonates and its impact on disease outcome. In humans, this can be done by the inclusion of a group of age-matched human infants with other types of cholestasis syndromes. Experimentally, this can be approached by bile duct ligation, but the inability to perform this type of surgery in 1–2-day-old mice makes this model not particularly suitable. In human BA, several viruses (e.g.: rotavirus, cytomegalovirus, reovirus) have been detected in different cohorts with the disease[32], but the lack of epidemiologic reproducibility or experimental evidence of direct causality in disease pathogenesis argues that other factors may contribute to tissue injury and the full phenotypic expression of the disease[33–36]. It is possible that the vertical transmission of butyrate or other metabolites via mother's milk contributes to disease protection in offspring. Although we did not measure the metabolites in nursing mice, concentrations of butyrate up to 750 μM have been found in breast milk and may modulate the expression of tolerogenic cytokines, the population of $T_{REG}$ cells, and anti-IL-1β activity[37]. Further, glutamate and glutamine are abundant free amino acids in human milk, with glutamine concentrations increasing to ~350% from 1 to 6 months of lactation[38].

Our findings identify the developing microbiome of neonates as one of the factors under the maternal influence that can modulate disease susceptibility in early postnatal life. By serving as a source of metabolites that can suppress immune cells (butyrate and glutamine) and promote survival signals in epithelial cells (glutamine), the microbiome can indirectly protect livers and bile ducts from virus-induced tissue injury. These data raise the potential for the use of butyrate and/or glutamine to modulate the immune response and aid tissue repair in BA or in other liver or biliary diseases of childhood. In addition to the disease-specific implications of our findings, the similarities in the microbial signatures between mothers and newborn mice and how they change upon dietary modifications during pregnancy underscore the maternal influence on key biological processes that may modulate the manifestation of clinical phenotypes in neonates.

## Methods

### Mice, experimental model of BA, and experimental procedures.
All animal studies were performed in strict accordance with the recommendations of the Institutional Animal Care and Use Committee of Cincinnati Children's Hospital Medical Center (Approved IACUC protocol: IACUC2020-0006; PI: Pranavkumar Shivakumar). Breeding pairs of adult BALB/c mice were obtained from Charles River Laboratories, acclimated, and maintained in microisolator cages in a specific pathogen-free facility. The mice were housed in a room equipped with a 12-h dark–light cycle and temperature and humidity were maintained between 65 and 75 °F and 40–60%, respectively. Mice had free access to water and sterilized chow (Purina LabChows Rodent Laboratory Chow 5010; Protein: 23.0%, Fat: 4.5%, Fiber: 6.0%) and were monitored daily by qualified veterinary staff to assure humane conditions. BA was induced in neonatal BALB/c mice within 24 h of birth by intraperitoneal injection with $1.5 \times 10^6$ ffu of RRV in a 20 μl volume; a similar volume of 0.9% saline solution was used as controls. Infected mice were monitored daily for jaundice, acholic stool, weight, and mortality until 2 weeks of life.

For butyrate- and propionate-based studies, mating pairs of adult BALB/c mice (8–10 weeks old) were fed ad libitum with free access to water alone or water + 200 mM sodium butyrate (Sigma-Aldrich, 303410) or water + 200 mM sodium propionate (Sigma-Aldrich, P1880) as reported previously[39–45]. The pH of butyrate and propionate solutions was adjusted to 7.5 using 0.1 N Hydrochloric acid. All dams and offspring were co-housed under the sodium butyrate or sodium propionate-treatment until weaning. Separate experiments used the oral administration of 0.3 mg of the butyrate or the intraperitoneal injection of 0.2 mg of inosine (Sigma-Aldrich, I4125) or 0.25 mg of L-glutamine (Sigma-Aldrich, G5792) per g of body weight of newborn mice initiated 1 day after RRV inoculation, followed by similar doses every other day until the time of sacrifice. For oral butyrate feeding in pups, we used a 25 μl Luer tip Hamilton syringe (Model 1702; Catalog # 80201) and a 26-gauge 1.25 mm ball diameter flexible oral gavage catheter (PetSurgical; Catalog # MDAFN2425S), following a previously published approach[8]. 100 mg of sodium butyrate was dissolved in 2 ml sterile water to give a final concentration of 50 μg/μl, pH adjusted to 7.5, and newborns were dosed daily at 0.3 mg/g body weight. The procedure involved holding the newborn between the thumb and index finger, gently inserting the catheter at a 45° angle, and administering the butyrate solution. Following gavage, the needle was gently withdrawn, and the pup was observed until normal activity and breathing patterns were regained.

### Histopathology and biochemical markers of cholestasis and tissue injury.
Livers and EHBD harvested from experimental cohorts of mice were fixed with 10% formalin for 24 h. Following fixation, tissues were embedded in paraffin, cut in longitudinal sections, and subjected to hematoxylin and eosin staining for histopathological analysis. Quantification of changes in the liver and bile duct histology was performed using previously published scoring systems[46,47]. At 12–14 days after saline or RRV inoculation, blood was collected by cardiac puncture and centrifuged at $3000 \times g$ for 6 min at 4 °C to obtain plasma. For blood sample collection, mice were selected randomly from separate litters to make sure the results were more representative of the much larger cohort. Total bilirubin and ALT were biochemically quantified in the plasma using Total Bilirubin Reagent Set (Pointe Scientific Inc., B7538-120) and DiscretPak ALT Reagent Kit (Catachem, V164-0A). Total bilirubin and ALT levels were colorimetrically analyzed using a Synergy H1 Hybrid Reader (BioTek) at 540 and 340 nm, respectively.

### Quantification of RRV titers.
RRV titers were quantified in tissue homogenates of livers and EHBD 7 days after infection of newborn mice from butyrate- and water-fed dams by fluorescent focus-forming units as previously described[48,49].

### Immune cell phenotyping.
Hepatic immune cell phenotyping was performed using hepatic MNCs obtained from livers of neonatal mice 7 days after challenge with saline or RRV[30,50,51], with the liver of each pup harvested, minced and dissociated using gentleMACS™ Dissociator (Miltenyi Biotec, 130-093-235). Flow cytometric markers used are as follows: $CD3^+CD4^+$ for $T_H$; $CD3^+CD8^+$ for $T_C$; $CD3^+CD4^+CD25^+Foxp3^+$ for $T_{REG}$; $CD3^+CD4^+CD25^+Foxp3^+IL-10^+$ for IL-10-expressing $T_{REG}$; $CD3^-CD49b^+$ for NK cells; $CD11b^+Gr-1^+$ for neutrophils; $CD11b^+F4/80^+$ for macrophages; $CD11b^-CD11c^+B220^+PDCA-1^+$ for plasmacytoid DCs (pDCs); $CD11b^+CD11c^+$ for myeloid DCs (mDCs) using the following flourochrome-conjugated monoclonal antibodies: FITC anti-mouse CD3 (clone: 17A2, eBioscience, 11-0032-82), Pacific Blue anti-mouse CD4 (clone: RM4-5, BioLegend, 100531), APC anti-mouse CD8a (clone: 53-6.7, eBioscience, 17-0081-81), PE/Cy7 anti-mouse CD25 (clone: 7D4, SouthernBiotech, 1595-17), PerCP/Cy5.5 anti-mouse CD3 (clone:17A2, BioLegend, 100218), FITC anti-mouse/rat Foxp3 (clone: FJK-16s, eBioscience, 11-5773-80), PE anti-mouse IL-10 (clone: JES5-16E3, eBioscience, 12-7101-41), APC anti-mouse CD49b (clone: DX5, BioLegend, 108910), PerCP/Cy5.5 anti-mouse CD11b (clone: M1/70, eBioscience, 101228), APC/Cy7 anti-mouse Gr-1 (clone: RB6-8C5, BioLegend, 108424), Pacific Blue anti-mouse F4/80 (clone: BM8, BioLegend, 123124), FITC anti-mouse CD11c (clone: N418, BioLegend, 117306), APC-eFluor® 780 anti-human/mouse B220 (clone: RA3-6B2, eBioscience, 47-0452-82) and APC anti-mouse PDCA-1 (clone: JF05-1C2.4.1, Miltenyi Biotec, 130-102-260). Intracellular staining for Foxp3 and IL-10 were performed using a Foxp3/Transcription Factor Staining Buffer Set (Life Technologies Corporation, 00-5523-00). The gating strategy for flow cytometric analyses is shown in Fig. S10A–K. The fluorescent signal was detected by

FACSCantoII dual-laser flow cytometer (BD Biosciences) and analyzed using the FlowJo software (Tree Star Inc.).

**Culture of RRV-primed hepatic MNCs with butyrate or fecal supernatant.** At 7 days after saline or RRV injection, the liver from each pup was harvested, minced, and dissociated using gentleMACS^TM Dissociator (Miltenyi Biotec, 130-093-235) separately to enable the quantification of individual cell types for each pup. The resulting cell suspension was filtered using a 40 µm nylon mesh (Fisher Scientific, 22363547). Hepatic MNCs were isolated using density-gradient centrifugation with 33% Percoll® (Sigma-Aldrich, P4937) followed by lysis of red blood cells using Ammonium-Chloride-Potassium Lysing Buffer (Gibco, A1049201). In all, $0.5 \times 10^6$ of the isolated MNCs were incubated with 0 (control), 0.1 or 0.25 mM of butyrate in 24-well flat-bottom plates. In separate experiments, 2-week-old neonatal mice from water- or butyrate-fed mothers were sacrificed, and colon contents were expressed from colon and cecum and homogenized using PBS (1 ml per 0.1 g of fecal matter). The diluted fecal contents were centrifuged at $12,000 \times g$ for 6 min at 4 °C and filtered using 0.22 µm syringe filters. The quantity of the supernatants was adjusted to 5% of the total volume to stimulate hepatic MNCs in vitro. After incubation at 37 °C for 3 days, total RNA was extracted using miRNeasy® Mini Kit (Qiagen, 217004), and genes were amplified using AriaMx Real-Time PCR System (Agilent Technologies, Santa Clara, CA) and target-specific primers to quantify gene expression levels for *Il10* (forward 5′-ATGCTGCCTGCTCTTACTGACTG-3′ and reverse 5′-CCCAAGTAACCCTTAAAGTCCTGC-3′), *Foxp3* (forward 5′-CACCCAAGGCTCAGAACTT-3′ and reverse 5′-GCAGGGGGTTCAAGGAA-GAA-3′), and *Tnfa* (forward 5′-ATGGCCTCCCTCTCATCAGT-3′ and reverse 5′-TTGGTGGTTTGCTACGACGT-3′).

**In vitro cytotoxicity assay.** Hepatic NK cells (effector cells) were purified from MNCs from 7-day-old mice with and without RRV challenge using MACS LS columns and mouse NK cell isolation kit (Miltenyi Biotec, 130-115-818). The purified NK cells were co-cultured with murine cholangiocyte cell line (mCL, target cells) in several target:effector cell ratios. The co-cultured cells (1:16 ratio of target to effector cells) were incubated with 3 ng/ml of recombinant mouse IL-10 (BioLegend, 575804) or increasing concentrations of glutamine: 4 (control), 8 and 16 mM for 5 h at 37 °C. To determine baseline cellular responses to glutamine, NK cells or mCL were separately pretreated with 16 mM of glutamine for 2 h, washed, and then co-cultured with control media for 3 h at 37 °C. To determine cell lysis in the presence of NAC, mCL was pretreated with 5 mM of NAC for 2 h, washed, and then co-cultured with NK cells for 3 h at 37 °C. In vitro cytotoxicity was determined by colorimetric measurement of lactate dehydrogenase released from mCL lysed by the NK cells using CytoTox 96 Non-Radioactive Cytotoxicity Assay (Promega, G1780) at 490 nm. The average absorbance value of the medium was subtracted from experimental as well as spontaneous control groups. The NK cell cytotoxicity was calculated as the percentage of cytotoxicity = (experimental − effector spontaneous − target spontaneous)/(target maximum − target spontaneous).

**Assay for GSH content.** GSH content was determined using a commercially available GSH assay kit (K261, BioVision, CA, USA). Liver slices and EHBD were obtained from day 12-old saline and RRV-challenged neonatal mice (with or without glutamine treatment) followed by homogenization in assay buffer with 5% sulfosalicylic acid (SSA). Tissue homogenates were centrifuged at $8000 \times g$ for 10 min to remove the protein and protein-free supernatants were assayed for GSH levels following the manufacturer's instructions. For cholangiocytes, cells were cultured in media containing different concentrations of L-glutamine (0, 4, 8, 16, and 32 mM) for 48 h and harvested at confluency in buffer containing 5% SSA. Cell-free supernatants were obtained by centrifugation of cell lysate at $700 \times g$ for 5 min and then assayed for total GSH content. In brief, specimens were incubated with a reaction mixture containing glutathione reductase and nicotinamide adenine dinucleotide phosphate oxidase for 10 min followed by GSH substrate (5,5'-dithiobis-(2-nitrobenzoic acid)) to generate 2-nitro-5-thiobenzoic acid. Total GSH (GSH + GSSG) was then measured spectroscopically at 415 nm. Protein content was determined using the Pierce BCA Protein assay kit (Thermo, Rockford, USA) to normalize the GSH levels.

**Statistical analysis for murine experiments.** Generalized linear mixed effect model with logit link was applied to binary responses of jaundice (Y/N), with repeated measurements associated with two variables, i.e., treatment groups (water or butyrate in maternal intake experiments, PBS or butyrate in oral administration experiments, and PBS, inosine, glutamine or inosine+glutamine in intraperitoneal administration experiments) and days after RRV-injection. Wald test was then applied to test the significant relationship between the logit of jaundice rate and each independent variable. For pairwise comparison of jaundice rates among the treatments across all study days, equality of coefficients from two different regression analyses was tested using Wald test with Bonferroni correction. Kaplan-Meier curves were generated to depict survival probability in the first 2 weeks of life, and Log-rank test with Bonferroni correction was used for comparisons of the survival curves. Flow cytometry, liver biochemistry (ALT and Bilirubin), in vitro fecal metabolite cell culture, and gene expression data with more than three groups were analyzed using analysis of variance (ANOVA) followed by Duncan's multiple comparisons while controlling for familywise error rates.

**NMR metabolomics.** Specimens for NMR-based metabolomics were prepared by filtering fecal water samples at $12,000 \times g$ for 90 min at 4 °C using prewashed 3 kDa NANOSEP 3 K spin filters (Pall Life Sciences, OD003C34) to remove macromolecules. NMR buffer containing 100 mM phosphate buffer in $D_2O$, pH 7.3, and 1.0 mM TMSP (3-trimethylsilyl 2,2,3,3-$d_4$ propionate) was added to 180 µL of fecal filtrate. The final sample volume was 220 µL and the final TMSP concentration in each sample was 0.182 mM. All experiments were conducted using 200 µL samples in 103.5 mm × 3 mm NMR tubes (Bruker, Z112272). One-dimensional $^1$H-nuclear Overhauser effect spectroscopy NMR spectra were acquired on a Bruker Avance II 600 MHz spectrometer using a noesygppr1d pulse sequence in the Bruker pulse sequence library. All free induction decays were subjected to an exponential line-broadening of 0.3 Hz. Upon Fourier transformation, each spectrum was manually phased, baseline corrected and referenced to the internal standard TMSP at 0.0 p.p.m. for polar samples using the Topspin 3.6 software (Bruker Analytik). For a representative sample, two-dimensional data, $^1$H-$^1$H total correlation spectroscopy, and $^1$H-$^{13}$C heteronuclear single quantum coherence were collected for assigning the metabolites. Chemical shifts were assigned to metabolites based on the reference spectra found in the Human Metabolome Database (HMDB)[52] and Chenomx® NMR Suite profiling software (Chenomx Inc. version 8.4). A total of 56 metabolites were quantified using Chenomx software based on the internal standard, TMSP. The metabolite concentrations were normalized to the original fecal sample weights, and the normalized values as mM/mg were used for statistical analysis.

**16s rRNA sequencing of murine stools and analysis of microbiome composition.** Colons were harvested from neonatal mice 12–14 days after saline or RRV inoculations and feces were collected from pregnant female mice between 1 and 2 days before delivery. The colons and feces were flash-frozen and stored at −80 °C until use. Total genomic DNA was extracted from the colons or feces using QIAamp DNA Stool Mini Kit (Qiagen, 51504). The concentration of the extracted DNA was measured with a Qubit dsDNA High Sensitivity Assay kit (ThermoFisher Scientific Inc., Q32854), and adjusted to 10 ng/ml in 15 µl volume. Quality control of the bacterial genomes in the extracted DNA was performed by amplifying V4 region of 16s rRNA with untailed 515F-primer: 5'-GTGCCAGCMGCCGCGG-TAA-3' and 806R-primer: 5'-GGACTACHVGGGTWTCTAAT-3', resulting in approximately 300 bp amplicon. For the gut-microbial sequencing, the V4 region was amplified by dual-indexed PCR amplification primers containing linker, i5 or i7 Nextera XT oligos (Integrated DNA Technologies), heterogeneity spacer, and V4 targeting primers[53]. The samples were run on Fragment Analyzer System (Agilent, M5310AA) with NGS Fragment Analysis kit (Agilent, DNF-473-0500), pooled using mass quantification, subjected to agarose gel electrophoresis, and purified with QIAquick Gel Extraction Kit (Qiagen, 28704). Ten pMolar of the library was used for next-generation sequencing with Illumina MiSeq platform and MiSeq Reagent Kits v2 (Illumina, MS-102-2003), resulting in de-multiplexed 250 bp paired-end reads in each direction[54]. The sequences were trimmed using Trimmomatic[55] to exclude reads with an average quality of less than 30.

We used a combination of best-of-breed analysis packages, including PANDAseq (PAired-eND Assembler for DNA sequences) v2.8[56], QIIME v1.8[57], and USEARCH v7.0.1090[58], along with custom scripts, and implemented a LONI (USC Laboratory of Neuro Imaging) pipeline workflow[59] for all 16 S rRNA preprocessing steps to obtain the OTU tables. The UPARSE-OTU algorithm at 97% of identity threshold was implemented with -cluster_otus option to construct a set of representative sequences per OTU. Global alignment -usearch_global was performed using forward strand nucleotides database -strand plus at 97% of identity threshold, -id 0.97. Qiime was used to generate a taxonomic assignment and phylogenetic reconstruction from the processed sequence data using python scripts as follow: align_seqs.py script to align the sequences to Greengenes database (v13_8) with NAST alignment algorithm, filter_alignment.py script to remove gap positions in the sequence, make_phylogeny.py script to produce phylogenetic tree from multiple sequence alignment by FastTree, and assign_taxonomy.py script to assign the taxonomy into each sequence with UCLUST algorithm. In addition, biom convert command was used to convert between BIOM and tab-delimited text files. The paired-end reads from each direction were assembled to single sequences resulting in 5,307,648 and 16,156,738 total reads, and 2,281,116 and 8,700,124 unique sequences after dereplication from the colons and feces, respectively. Sort-by-size, OTU clustering and mapping reads to OTUs resulted in 351 and 2,469 OTU representative sequences from the colons and feces, respectively.

Fisher's exact test was used to examine if the number of OTUs between pregnant females and neonatal mice is associated with BA-phenotypes in $2 \times 2$ contingency tables. Chi-square test was employed to examine whether the composition of bacterial phylum in shared OTUs differs from that in the unique bacterial population. Adjusted standardized residuals were calculated to determine the observations that significantly deviated from expectations in each contingency table. For beta diversity (between groups), NMDS was used to ordinate the microbial communities or metabolite profiles based upon Euclidean distance. Analysis of similarity was used to test the statistical significance for pairwise comparisons in the NMDS plots. Significantly different abundance of taxa between diseased and resistant neonatal mice was identified by Kruskal–Wallis sum-rank test and linear discriminant analysis (LDA) effect size with 3.5 of threshold on the logarithmic score of LDA analysis[60]. The significantly different abundance of taxa

between diseased and resistant neonatal mice was visualized in circular cladograms generated by GraPhlAn[61]. Statistical analyses were performed with the STATISTICA 7 (StatSoft, Tulsa, OK), Prism 8 (GraphPad Software, San Diego, CA), SAS 9.3 (SAS Institute, Cary, NC), and PAST 3 softwares[62].

**Human study and stool sample collection**. The multicenter study followed the ethical guidelines of the 1975 Declaration of Helsinki as reflected in the approval by the institutional review board and ethics committee at each participating center. Consent and agreement were obtained from parents or legal guardians to participate in the research. Study protocols were approved by the institutional review board (IRB) and ethics review committee of the following institutions/centers: Department of Pediatric Surgery, Union Hospital, Tongji Medical College, Huazhong University of Science and Technology, Wuhan, Hubei, 430022, China; Department of Neonatal Surgery, Xi'an Children's Hospital, Xi'an, Shanxi, 710003, China; Department of Pediatric Surgery, Wuhan Children's Hospital, Tongji Medical College, Huazhong University of Science and Technology, Wuhan, Hubei, 430015, China; Department of General Surgery, Shenzhen Children's Hospital, Shenzhen, Guangdong, 518038, China; Department of Pediatric General Thoracic and Urology Surgery, The Affiliated Hospital of Zunyi Medical University, Zunyi, Guizhou, 563000, China; Department of Pediatric Surgery, Jiangmen Maternity and Child Health Care Hospital, Jiangmen, Guangdong, 529000, China. Subjects with BA were enrolled in a non-interventional prospective study from November 2017 until July 2019 at the Pediatric Care Centers in the Chinese cities of Wuhan, Xi' An, Shenzhen, Zunyi, and Jiangmen; age-matched participants without liver diseases were also enrolled to serve as controls. Stools were collected at the time of clinical diagnosis prior to surgery. The demographic characteristics including age, sex, diet, and liver biochemical parameters are included in Table S6. The inclusion criteria required confirmative diagnosis of BA by intraoperative cholangiography and demonstration of fibrosing obstruction of EHBD. The selection criteria for age-matched controls included age between 2 weeks and 6 months, no fever, no liver disease, no digestive symptoms such as constipation and diarrhea at the time of stool collection, and no antibiotic administration within 2 weeks.

Fecal samples were collected following the instructions of OMNIgene•GUT (OMR-200), immediately after defecation. A total of 129 stool specimens from BA patients and 32 controls were submitted for sequencing. After initial quality control for microbiome integrity and quantity, 102 BA and 28 control specimens met quality criteria for subsequent library construction and were subjected to metagenomic sequencing (Fig. S9).

**Human fecal genomic DNA extraction and shotgun metagenomic sequencing**. The microbial community DNA was extracted using MagPure Stool DNA KF Kit (Angen Biotech, MD5115). DNA was quantified with a Qubit Fluorometer by using Qubit dsDNA BR Assay kit (ThermoFisher Scientific, Q32850) and the quality was checked by running an aliquot on 1% agarose gel. After DNA extraction, 1 μg genomic DNA was randomly fragmented by Covaris, followed by purification using the AxyPrep Mag PCR clean-up kit (Axygen, MAG-PCR-CL-250). The fragmented DNA was selected using Agencourt AMPure XP PCR Purification Kit (Beckman Coulter, A63881) to an average size of 200–400 bp. The fragments were end-repaired by End Repair Mix and purified afterward. The repaired DNAs were combined with A-Tailing Mix, then the Illumina adapters were ligated to the Adenylate 3'Ends DNA and followed by purification. The products were selected based on the insert size. Several rounds of PCR amplification with PCR Primer Cocktail and PCR Master Mix were performed to enrich the Adapter-ligated DNA fragments. After purification, the library was qualified by the Agilent 2100 bioanalyzer and ABI StepOnePlus Real-time PCR System. Finally, the qualified libraries were sequenced on Illumina Novaseq 6000 platform (BGI-Shenzhen, China). Each sample that passed the initial quality check, was run in 2 × 150 bp paired-end reads, producing a median of 43 (IQR: 37–44) million reads per sample. The clean data output is above 10 Gb per sample.

**Human fecal microbiome analysis**. Tools from the bioBakery meta'omics analysis environment were used for microbial community taxonomic and functional profiling[63]. Initial quality control was performed using KneadData v0.7.3. Reads mapping to the human genome and those <90 bp after truncation at the first four base sliding window with an average phred score <20 were filtered prior to profiling. Taxonomic profiling with virus detection was performed using MetaPhlAn2 v2.9.5[64] with the default parameters and v2.9.5 CHOCOPhlAn database to obtain microbial relative abundances for each sample. The relative abundance of gene families was identified by mapping reads to the UniRef90 protein reference database (https://academic.oup.com/bioinformatics/article/31/6/926/214968). Multi-sample community profiles and clinical metadata files were integrated into phyloseq v1.28.0 objects[65] for statistical analysis. Differentially abundant microbial species in samples from BA and control patients were identified at each taxonomic rank from phylum to species using moderated negative binomial regression as implemented by DESeq2 v1.24.0[66]. Estimated read counts were obtained by multiplying the species-level microbial relative abundance by the total read count after filtering of contaminant and low-quality sequences. Organisms not seen in at least 20% of samples were filtered prior to analysis. Normalization was performed using the sftype = "poscounts" and local fitting specified using the fitType = "local"

commands in DESeq2. Taxa with a Benjamini-Hochberg false discovery rate (FDR) corrected $p$ value of <0.05 were considered as differentially abundant. The number of observed species and Shannon diversity were calculated using the specnumber and diversity functions in vegan v2.5.5[67] with differences between BA and control samples tested using Wilcoxon rank-sum tests. Non-metric dimensional scaling was performed on the Bray-Curtis dissimilarity matrix and Jaccard distance to assess the clustering of samples according to sample type.

Functional profiling of gene families and pathways was performed using HUMAnN2 v2.8.1[68] with the default parameters. Pathway abundances were obtained by aggregation of gene families to MetaCyc pathways[69]. Differentially abundant MetaCyc pathways were identified using the ANOVA-Like Differential Expression (ALDEX2) package v1.16.0[70] for high-throughput sequencing data. Pathways not seen in at least 20% of samples were filtered prior to analysis and counts per million transformed to interquartile log-ratios prior to analysis. Tests for differential abundances were conducted using the Wilcoxon rank-sum test averaged over 128 Monte-Carlo instances drawn from the Dirichlet distribution. Pathways with an FDR corrected $p$ value of <0.05 were considered as differentially abundant. All analyses were performed using the R software environment for statistical computing and graphics v3.6.0[71].

**Integration of predicted functional pathways and bacterial taxa**. To measure the strength and direction of the association between significant butyrate/butanoate pathways and bacterial taxa from experimental BA, Spearman's correlation coefficients, $p$ values, and correlogram were obtained for differentially abundant taxa and MetaCyc pathways using the *rstatix* module in the R package: cor_mat(method="spearman"), cor_get_pval(), and cor_plot(method="color",insignificant="blank"), respectively. A unique set of bacterial butyrate synthesis genes were obtained by combining the gene lists from the individual butyrate pathways from MetaCyc. Species-level contributions to differentially abundant MetaCyc pathways in BA and NC samples were obtained using HUMAnN2. Butyrate-specific enzymes and genes for bacterial species enriched in NC subjects were inferred from the BioCyc database (https://biocyc.org).

**Reporting summary**. Further information on research design is available in the Nature Research Reporting Summary linked to this article.

## Data availability

The 16s rRNA sequencing files and metadata of experimental biliary atresia have been deposited in the European Nucleotide Archive (ENA) at EMBL-EBI and are available under accession number PRJEB40649. The raw NMR spectra files of experimental biliary atresia are accessible from MetaboLights (Study Identifier: MTBLS2171, https://www.ebi.ac.uk/metabolights/reviewerabff7e6f-abf1-44cb-976a-9fd380ee712b). Data underlying all figures are provided as a Source data file in Excel format. The raw metagenomic sequencing reads of human stools and host-phenotype meta-data used in this study are deposited and accessible in the European Genome-phenome Archive data repository with accession code "EGAD00001007735". Due to participant confidentiality and informed consent agreements, the raw sequencing files are available upon request to the data access committee of this collaborative study. The data access committee consists of research collaborators from Cincinnati Children's Hospital Center and Tongji Medical College, Huazhong University of Science and Technology. Letter of intent can be submitted to the following primary contacts for restricted access to human sequencing data: S.-t.T., tshaotao83@hust.edu.cn. and J.A.B., Jorge.Bezerra@cchmc.org. The letter of intent shall expect a response from the data access committee within 2–4 weeks. Data access is subject to local rules and regulations. Detailed data use agreement will be shared after the letter of intent is received, reviewed, and approved by the data access committee. All other data related to this article are included in the Supplementary Materials and Supplementary Tables and are available to the readers. Other information is available from the corresponding authors upon reasonable requests. Source data are provided with this paper.

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

## Acknowledgements

We thank the Clinical Component, the Gene Analysis Core, and the Integrative Morphology Core of the Digestive Health Center at Cincinnati Children's Hospital Medical Center. This waork is supported by NIH research project grants DK 64008, DK 83781, DK 78392 (to J.A.B.) and by the National Science Foundation of China grant # 81670511 (to S.-t.T.). Neither J.A.B. nor other co-authors from Cincinnati Children's Hospital Medical Center have secondary appointments or receive funds from collaborating Chinese institutions.

## Author contributions

J.J.J., L.Y., and P.S.: conceptualization, data generation and curation, formal analysis, investigation, methodology, project administration, software, validation, visualization, writing manuscript; P.-p.X.: data curation, investigation, project administration; R.M. and U.T.: methodology, project administration; P.Y., Y.Z, Y.P., H.W. X.D. Y.Y., B.W., Z.J., Y.L., and Z.C.: data curation, methodology, and resources; M.W.-C., M.W., N.J.O., Z.L., and L.F.: data curation, formal analysis, resources, software; L.E.R.-R.: resources and data generation; S.-t.T.: conceptualization, funding acquisition, supervision; J.A.B.: conceptualization, funding acquisition, data analysis, methodology, project administration, supervision, writing manuscript. All authors contributed to drafting, reviewing, and editing the manuscript.

## Competing interests

The authors declare no competing interests.
