## [Peer Review File · Nature Communications]

REVIEWER COMMENTS

Reviewer #1 (Remarks to the Author):

To the authors:

The original research manuscript entitled “Maternal regulation of biliary disease in neonates via gut microbial metabolites”, shows the microbiome impact of oral butyrate and glutamine in newborns mice and their mothers with biliary atresia caused by a rotavirus infection. Their findings suggested that butyrate and specially glutamine have a profound impact in suppressing activated lymphocytes and that glutamine showed resistance of cholangiocytes to NK- induced lysis. Furthermore, metagenomic analysis of neonate with biliary atresia and healthy were compared in order to validate the findings from the animal experiments.

General comments: The manuscript is well written; the design of the project, but some of the experiments are not fully supported by the findings. Moreover, all experiments were very cleverly carried out. The authors demonstrate the importance and implication of butyrate and glutamine in the neonate’s disease. The manuscript shows very novel findings, which I consider, are of high impact for the scientific community especially for gastroenterologist. However, there are few major concerns and several minor comments that overshadow the work and authors should be address.

Major comments:

1. The abstract is one of the most important parts of an article as it is the only text that for sure all the readers will look at, before deciding to download/open the full paper. To my personal opinion, it is incomplete, I recommend adding the human experiments. These are highly important because they are made in current clinical samples.
2. It is important to explain before showing results in Fig 1, the difference between “Disease” and “Resistant”, in the butyrate + RRV groups and explain about the ratio between groups.
3. In order to make it easier for the reader, include each sub-figure in the corresponding place.
4. I strongly suggest inserting small experimental design diagrams, in order to have clear idea of the conditions of the experiments in animals.

5. Analysis of fecal metabolites in neonatal mice from butyrate or water feed-mothers showed hypoxanthine and glutamate as the most significant signatures increased by butyrate. On the other hand, the metagenomic analysis in human neonates showed “urea cycle” and “citrulline biosynthesis” functional pathways as the most significant. However, from both results (one obtained in mice and other in humans, and one obtained by metabolomics and another by metagenomics), you can conclude that Hypoxanthine and Glutamine/glutamate are the key metabolites to test. Indeed, “urea cycle” does not seem to include glutamine neither glutamate. And how glutamine was selected? As Glutamine in Fig 3F is one of the lowest importance in the Euclidean distance. Later on, glutamate is forgotten and experiments are focused only in glutamine.

6. To my personal opinion, organisms can not be compared like that in different conditions, as the mice were not showing biliary disease and the omics applied are not the same, genes and metabolites can not be compared like that. What about doing metabolomics in a subset of human samples? To reach better conclusions?

7. Why do you think that in humans neonates Shannon index and NMDS plots were not so different between groups.

8. How you change glutamine/glutamate to only select glutamine?

9. As before, why inosine was selected instead of hypoxanthine?

10. In my personal opinion, inosine and glutamine were selected from literature knowledge, as examples please look at these publications:

1. Kroemer, G., Zitvogel, L. Inosine: novel microbiota-derived immunostimulatory metabolite. *Cell Res* (2020). <https://doi.org/10.1038/s41422-020-00417-1>

2. Wu, M., Xiao, H., Liu, G., Chen, S., Tan, B., Ren, W., Bazer, F.W., Wu, G. and Yin, Y. (2016), Glutamine promotes intestinal SIgA secretion through intestinal microbiota and IL-13. *Mol. Nutr. Food Res.*, 60: 1637-1648. doi:10.1002/mnfr.201600026

11. Please explain the results obtained in bilirubin and ALT graphs for the Inosine+glutamine group. Why their results are different from glutamine alone.

12. Lines 184, “While pre-incubation of NK cells with glutamine did not suppress cell lysis,” please review the sentence, as it was the incubation with IL-10 where cell lysis was not suppressed.

13. Discussion. Please include references of other publications that help to support your results. Moreover, it is very important to stick to your findings. For example (line 196-197): “we found similar signatures of fecal microbiome in mother and newborns, the enrichment of butyrate and glutamine in the newborns’ fecal metabolites,” fecal metabolites were from newborn mice (from mothers feed by butyrate or water).

14. Line 211-213. Another example from the above idea is: “A stronger cue to the relevance of the microbial population was the increase of glutamine in fecal metabolites of newborn mice resistant to

biliary injury.”, glutamine was indeed increased, but was not one of the strongest and did not come from newborn mice resistant to biliary injury but from mothers fed with butyrate only.

15. Statistics. For all figures, when you have more than 2 groups, an ANOVA or Kruskal-wallis test must be performed. For using ANOVA, data should follow normality, so normality should be tested. Later on, pair-wise tests can be applied between groups, and p value will be corrected.

16. Please check the reference, for the main manuscript only 5 were used, while for M&M 23. If it necessary, include more as the consideration of the authors.

17. Why experiments from Figure 4P-Q were done only for glutamine and not for butyrate?

Minor comments:

1. Line 77, please add the reference of the previous publication.

2. Line 88, PBS is control group or water? Please unify to make it clear.

3. Line 94, “extrahepatic bile ducts of neonates from butyrate-fed mothers were patent”, please check, the sentence looks odd.

4. Line 123-125, “rotavirus-infected mice that did not develop biliary obstruction” please include the name of the groups such as “butyrate-RRV resistant” for example. So that the groups in the figure match with the sentence in the text.

5. Line 127, delete female, pregnant are always female.

6. Line 128-129, “predominant enrichment of Firmicutes and Bacteroidetes with a decrease in Proteobacteria”, please review this sentence, proteobacteria seems to be decrease when diseased and resistant are compared.

7. Line 127, what about the unique 1315 OUT’s unique from the pregnant mice? These are almost 70% of the total OUT’s from the mothers, and mothers and disease-resistant babies only share 30% of their mothers.

8. Please be consist through the text, as use IL or II for interleukin, but not both.

9. Please change rpm units by xg, otherwise your experiments can not be reproducible for other research groups.

10. Line 252, you have an underscore after a dot.

11. Please define mCL

12. Figures footnotes, for mean +/- SD, it does not look to be SD, but to be SEM, please check.
13. Lines 478 and 480, please add the reference number of the data submitted to the electronic libraries.
14. For all figures, in the immunohistochemical pictures, please add the scale.
15. Figure S1. Line 26 "show increased TREG and IL-10+TREG cells", these type of line are results, they should not be included as part of the figure.

Reviewer #2 (Remarks to the Author):

This well written manuscript explores the significance of the maternal and pup/infant fecal microbiome in biliary atresia (BA). The authors show that feeding butyrate to pregnant mice reduces the susceptibility of infant mice to RRV-induced BA. Fecal microbiome signatures in the mice pups were similar to those in infants with BA, and fecal metabolomic analysis showed enrichment with both butyrate and glutamine. Both suppressed activated lymphocytes, however only glutamine protected cholangiocytes in culture from NK cell lysis. The authors conclude that fecal metabolite composition in infants may modulate injury and BA disease expression in humans. This study raises a number of methodologic and other questions:

1. The authors show that feeding mouse dams butyrate or feeding the mouse pups butyrate protects the pups from RRV-induced BA. It is quite possible that the reason for this is a protection against RRV infecting cholangiocytes or reducing the duration of this infection. This was not evaluated. It would be important for the authors to perform plaque assays following RRV infection to determine if the frequency of active infection and the duration of infection was similar or different in the mice from butyrate fed dams vs. controls.
2. Did all of the mice in each of the litters from each of the butyrate fed dams respond similarly to protection from BA? This would be expected since each fetal mouse should have been exposed to the same intrauterine environment as well as maternal fecal microbiome after birth.
3. It is assumed that the intrahepatic cell isolations in the mouse pups was from pooled livers. The authors should specify the pooling strategy that was used.
4. How was butyrate administered "orally" to RRV infected mice on day 2 of life and thereafter?
5. The decreased IL-10 from hepatic mononuclear cells from RRV infected pups that were incubated with butyrate is not consistent with the statement in the abstract that butyrate added to lymphocytes suppresses their activity. This should be corrected in the abstract.

6. A relatively small number of animals had liver biochemistries measured compared to the total number in Figure 1A. Were liver biochemistries performed on random mice from separate litters to make the results (n=5-7) more representative of the much larger cohort (n=58-72) that were studied?

7. How do the authors explain that the hypoxanthine in stool appeared to be protective in the BA mice but was increased in stool of BA human infants?

8. In several figures, liver and bile duct histology from one animal in each group is shown. How representative was this? How many animals had liver histology and bile duct histology measured for each of these Figures, and what percent showed similar reductions in inflammation, obstruction, etc.? How did the authors quantitate changes in liver histology (scoring system)?

9. The cholangiocyte protective effect of glutamine when incubated with NK cells was quite interesting. Since glutamine is one of the precursor amino acids of glutathione, did the authors measure glutathione levels in the cholangiocytes following glutamine incubation? Did glutathione levels increase in the liver of glutamine supplemented rat pups vs. the controls or the inosine supplemented pups? There are previous studies suggesting that glutathione and its pathways may be low in liver of human infants with BA, so this should be examined in these experiments and discussed in the Discussion.

10. Were any other amino acids used as controls in the incubation experiments of glutamine and butyrate with measurement of cholangiocyte lysis? It would be interest to use N-acetyl cysteine in these experiments as another means of increasing glutathione levels to determine if this would protect against NK-induced cholangiocyte lysis.

Reviewer #3 (Remarks to the Author):

The manuscript by Jee J et al investigates the effects of butyrate and glutamine on susceptibility to biliary injury. The authors hypothesized that mothers regulate the resistance to this disease via the intestinal microbiome transmission to offspring. A high butyrate diet to mice and their offspring conferred resistance to biliary injury. Similarities in microbiome signatures shared between mothers and offspring were used to postulate a microbiota-mediated protection. This was followed by a metabolomics analysis, revealing glutamate and hypoxanthine as the most differing between butyrate fed and water controls. Microbiome composition as well as pathways analyses from healthy humans and patients suffering from biliary disease indicated different microbiome compositions of these two groups. Finally, Glutamine and Inosine were administered to infected mice and disease markers/immune signatures were analyzed as above.

The impact of the maternal and early life microbiomes on disease trajectories is of great significance. The effects of butyrate and glutamine administration are interesting and appear convincing (though I am not an expert on such analyses). My main issue is that the mechanistic microbiota-metabolite link seems less well established. Therefore, some conclusions maybe overarching on the basis of the observed associations.

Major points

- An important concern is the physiological relevance and impact of orally administering high butyrate concentrations. The authors are suggesting the conclusions from this work resemble the case of bacterially produced butyrate in the colon (see lines 217-218, 226-228). This is problematic as there are key differences, e.g. Naturally, butyrate production is not occurring in the small intestine, but in the colon by mucin adherent bacteria. Butyrate is also not a part of human diet, so small intestinal microbiota are unlikely to be exposed to butyrate in natural settings, much less at such concentrations. This artificial butyrate administration per se may affect the microbiota, and thereby human health, e.g. by inhibiting/stimulating microbial groups. What is the justification of using 200 mM butyrate? This high concentration is likely to affect the pH and have some impact on the microbiota of the small intestine and the colon. The authors are advised to present and discuss a comparison of the microbiota of the healthy mice controls to butyrate fed animals to highlight potential differences. Such differences due to the unnaturally high butyrate may not resemble changes expected by promoting natural bacterial butyrate production in the colon, e.g. through dietary supplements? The amount and the localisation of butyrate are likely to be of significance to its physiological impact?. A justification of why authors have used 0.1-0.25 mM butyrate additions in cultures hepatic mononuclear cell culture should be mentioned, given the large difference to what is administered, i.e. do the authors know what concentration of butyrate makes it through the PV to the liver? Would be nice if the authors address these issues and maybe briefly discuss them in the manuscript.

- The presentation of the microbiome analyses is not sufficiently clear to warrant the simple conclusion that maternal butyrate-confers maternal producing microbiome in neonates. It does not appear that the changes in microbiota are limited to butyrate producing bacteria, but these changes seem broad, generally affecting colonic versus small intestinal microbiota. Thus, a higher relative abundance of small intestinal members, e.g. bacilli and Proteobacteria are associated with disease, whereas the relative abundance of colonic Bacteroidetes, clostridial Firmicutes and Actinobacteria is lower. Can the authors present any evidence that these changes are the cause of the disease rather than a manifestation of it?

If not, then the text needs to be nuanced and toned down to discuss both scenarios. Maybe also the microbial profiles of the diseases vs resistant should be shown in addition to the subsets shared by different as shown in Fig. 2F (so just a comparison of the profiles of all identified OUT). What about the unique taxonomic groups for resistant vs diseased, maybe those also should be presented at least in the SI and discussed if relevant. Orthogonal transfer of microbiota is well established, and indeed the effects observed in mice, could well be microbiota mediated. However, can the authors

preclude that butyrate or other metabolites that confer resistance are transferred to the offspring via mother's milk? Perhaps this can be discussed as a potential route to confer health effect on the offspring, unless the authors have measured metabolites in the milk of nursing mice.

- The choice to administer Gln rather than Glu and Inosine rather than hypoxanthine is puzzling. The metabolite analyses showed that Hypoxanthine and Glu are the most differentially abundant in resistant mice, whereas changes in Gln were modest (see Fig. 3E,F). Therefore, this choice needs to be justified. Accordingly, the mentioning of Gln enrichment in the abstract appears subjective, and does not accurately reflect the data (enrichment of Glu rather than Gln). Also I am not sure what the performed pathway analysis is adding as it has been presented. The presence of genes is not per se informative on their expression level and the identified pathways are central and both Glu and Gln are central metabolic precursors. Expression data would have been more suitable to make claims on this, otherwise, the authors should attempt to see if the "enriched pathways" they describe are somehow associated with the taxonomic groups identified as signatures for resistance. A distinction needs to be made between Glu and Gln also in the text. For example, the urea cycle in bacteria involves only Glu to my knowledge and not Gln. Obviously Glu can be generated from Gln and vice versa, but this has not been specifically addressed, e.g. by analysis of the abundance of the genes involving the interconversion, i.e. glutamine synthetase or phosphate dependant glutaminase. So it is warranted to analyse/discuss this to make the transition from Glu to Gln logical. The Urea cycle in some G-positive Bacilliales has been proposed to be activated in response to acidic stress, (see 10.1128/JB.02517-14), which maybe be interesting in the context of pH changes in the small intestine related to biliary injury?. Also, the initial hypothesis on butyrate, maybe should be evaluated by looking at the enrichment of butyrate synthesis genes in the cohort.

Minor:

- P7, L144 the "...greatest differential expression" is not appropriate, this is abundance and not expression you are measuring.
- Line 252, there is an issue with a funny dash.

RESPONSE TO REVIEWERS

The comments and questions from the reviewers were very insightful. To address them, we performed new data analysis, carried out additional experiments, and revised the texts and figures. By addressing the reviewers' comments and questions, we improved the manuscript, as discussed below in the point-by-point response. All changes in the manuscripts are shown in red to facilitate the re-review of the manuscript.

Manuscript #

NCOMMS-20-36817

Manuscript Title

Maternal regulation of biliary disease in neonates via gut microbial metabolites

REVIEWER #1

Major comments

1. *The abstract is one of the most important parts of an article... I recommend adding the human experiments.*

Reply: Following the reviewer's suggestion, we revised the manuscript to include the human data in the abstract. The abstract now reads: "...In human neonates with biliary atresia, the fecal microbiome was enriched with Proteobacteria and Bacilli and had suppressed metabolic pathways for glutamate/glutamine at the time of diagnosis." (see Page 3).

2. *It is important to explain before showing results in Fig 1, the difference between "Disease" and "Resistant", in the butyrate + RRV groups and explain about the ratio between groups:*

Reply: We revised the manuscript to clearly explain the differences. Page 5 now reads: "...the clinical evolution of mice from butyrate-fed mother segregated into a group of mice with a similar course of progressive jaundice and mortality (40% of pups, referred to as "Diseased") and a second group that showed only transient jaundice and survival (60% of pups, referred to as "Resistant")..." These two phenotypes are also clearly identified in drawings now included in the revised Fig. 1.

3. *In order to make it easier for the reader, include each sub-figure in the corresponding place... I strongly suggest inserting small experimental design diagrams, in order to have clear idea of the conditions of the experiments in animals.*

Reply: In response to this helpful suggestion, we now include each subfigure in the corresponding place and diagrams depicting experimental designs for each figure to give the reader a clear idea of the logical sequence of the experiments, intervention and control groups and subgroups, and the data being presented. Thank you for the suggestion.

4. *Analysis of fecal metabolites in neonatal mice from butyrate or water fed-mothers showed hypoxanthine and glutamate as the most significant signatures increased by butyrate. On the other hand, the metagenomic analysis in human neonates showed "urea cycle" and "citrulline biosynthesis" functional pathways as the most significant.*

However, from both results (one obtained in mice and other in humans, and one obtained by metabolomics and another by metagenomics), you can conclude that Hypoxanthine and Glutamine/glutamate are the key metabolites to test. Indeed, “urea cycle” does not seem to include glutamine neither glutamate. And how glutamine was selected? As Glutamine in Fig 3F is one of the lowest importance in the Euclidean distance. Later on, glutamate is forgotten and experiments are focused only in glutamine.

Reply: The reviewer is correct that glutamate ranks higher in Euclidean distance in relation to glutamine and that hypoxanthine and glutamine/glutamate are key metabolites to test. In regard to glutamine/glutamate and the urea cycle, they are commonly placed into the urea cycle pathway analysis due to its role in ammonia (NH₃) formation upon cleavage to form glutamate and NH₃, which reacts with ATP and bicarbonate ion to form carbamoyl phosphate (CP), a key intermediate in the Urea Cycle. Our rationale for choosing glutamine is based on a previous report of low glutamine levels in plasma of children with biliary atresia (Reference: Zhao D, et al. PLoS One 2014; e85694) and on three additional features of glutamine: 1) its role in influencing intracellular bio-energy utilization, whereas glutamate (with its negative charge) has specific biological functions such as in neurotransmission; 2) the described role of glutamine in modulating Treg cell function and ox-redox balance via GSH (two important mechanisms of pathogenesis in biliary atresia); and 3) the safety profile of glutamine with a long-history of use in parenteral nutrition in humans. In response to the reviewer’s helpful comments, we revised the manuscript to include these considerations and supporting references (Pages 9–10).

5. *To my personal opinion, organisms cannot be compared like that in different conditions, as the mice were not showing biliary disease and the omics applied are not the same, genes and metabolites cannot be compared like that. What about doing metabolomics in a subset of human samples?*

Reply: It would be ideal to perform metabolomics in a subset of human samples, but the collection of fecal specimens used protocols that optimizes bacterial DNA integrity, thus making the specimens not suitable for metabolomic analyses. We filled this gap in human metabolomics data by an effective analytical genomics data strategy that identified a convergence of metabolites and pathway enrichments to hypoxanthine and glutamate/glutamine. We respectfully propose that our ability to directly test these metabolites in several murine and cell culture experiments validates, at least in part, the omics analysis and findings.

6. *Why do you think that in humans neonates Shannon index and NMDS plots were not so different between groups?*

Reply: The lack of differences in the Shannon index and NMDS plots may relate to the inherently low diversity of the intestinal microbiome in children (Reference #13: Radjabzadeh EG et al. Sci Rep 2020;10:1040), which may be particularly underdeveloped in younger infants. This is mentioned on Page 8 of the revised manuscript. This is one finding that may trigger future investigation by us or other investigators in the field.

7. *How you change glutamine/glutamate to only select glutamine?*

Reply: As discussed above, we chose glutamine based on a previous report of low glutamine levels in plasma of children with biliary atresia (Reference #18: Zhao D, et al. PLoS One 2014; e85694) and on three additional features of glutamine: 1) its role in influencing intracellular bio-energy utilization, whereas glutamate (with its negative charge) has specific biological functions such as in neurotransmission; 2) the described role of glutamine in modulating Treg cell function and ox-redox balance via GSH (two important mechanisms of pathogenesis in biliary atresia); and 3) the safety profile of glutamine with a long-history of use in parenteral nutrition in humans - See edits on Pages 9–10 of the revised manuscript.

8. *As before, why inosine was selected instead of hypoxanthine?*

Reply: The main reasons for the use of inosine include: 1) hypoxanthine is poorly soluble in water, cannot be administered parenterally, and not intended for clinical use due to its potential toxicity; 2) inosine serves as a precursor of hypoxanthine; and 3) inosine has immunoregulatory properties (biological processes that have been previously implicated in pathogenesis of biliary atresia). We have revised the manuscript to more clearly show this rationale (Page 10).

9. *In my personal opinion, inosine and glutamine were selected from literature knowledge, as examples please look at these publications: a) Kroemer, G., Zitvogel, L. Inosine: novel microbiota-derived immunostimulatory metabolite. Cell Res (2020). <https://doi.org/10.1038/s41422-020-00417-1> and b) Wu, M., Xiao, H., Liu, G., Chen, S., Tan, B., Ren, W., Bazer, F.W., Wu, G. and Yin, Y. (2016), Glutamine promotes intestinal SIgA secretion through intestinal microbiota and IL-13. Mol. Nutr. Food Res., 60: 1637-1648. doi:10.1002/mnfr.201600026.*

Reply: This is a remarkably insightful comment, which we found them very helpful. Both references are important in the field and are now part of the revised manuscript (References #14 and #17).

10. *Please explain the results obtained in bilirubin and ALT graphs for the inosine + glutamine group. Why their results are different from glutamine alone.*

Reply: The main reason for the increases in bilirubin and ALT in mice receiving inosine+glutamine is the presence of liver injury as shown in Fig. 6O (depicting hepatitis and portal inflammation). Careful histological analysis shows an improved epithelial lining of extrahepatic bile ducts (like it is seen in mice receiving glutamine alone), but an unexpected ongoing presence of hepatitis and portal inflammation. This may be secondary to a liver-specific toxicity of inosine in the setting of a viral infection or to different biological processes regulating the hepatic vs extrahepatic bile duct injury.

11. *Lines 184, “While pre-incubation of NK cells with glutamine did not suppress cell lysis,” please review the sentence, as it was the incubation with IL-10 where cell lysis was not suppressed.*

Reply: We examined the paragraph, this sentence, and our data on Fig. 7. The reviewer is correct that the addition of IL-10 (Fig. 7D) did not suppress cell lysis. It is also true that pre-incubation of NK cells alone with glutamine did not suppress cell lysis (Fig. 7F). Thus, neither the presence of IL-10 nor the pre-incubation of NK cells with glutamine prevented cholangiocyte lysis. We edited the sentences to make sure we the correctly present the results in the manuscript.

12. *Discussion. Please include references of other publications that help to support your results. Moreover, it is very important to stick to your findings. For example (line 196-197): “we found similar signatures of fecal microbiome in mother and newborns, the enrichment of butyrate and glutamine in the newborns’ fecal metabolites,” fecal metabolites were from newborn mice (from mothers feed by butyrate or water).*

Reply: We have revised the manuscript to incorporate the reviewer’s suggestions. Specifically, we now include new references that help support our results (ref. #1-4, #17-23, #28, and #36-38). In regard to the statement on lines 196–197, we agree with the reviewer that the fecal metabolites were from neonatal mice born to butyrate- or water-fed mothers. The revised text reads: “...similarly to the fecal microbial signature of their butyrate-fed mothers...” (Page 12).

13. *Line 211-213. Another example from the above idea is: “A stronger cue to the relevance of the microbial population was the increase of glutamine in fecal metabolites of newborn mice resistant to biliary injury”, glutamine was indeed increased, but was not one of the strongest and did not come from newborn mice resistant to biliary injury but from mothers feed with butyrate only.*

Reply: We agree with the reviewer’s assessment. We have revised the manuscript to state that “...glutamate and glutamine in fecal metabolites of neonatal mice born from butyrate-fed mothers and the findings that glutamine administration largely prevented the development of experimental biliary atresia in neonatal mice...” (Page 12).

14. *Statistics. For all figures, when you have more than 2 groups, an ANOVA or Kruskal-wallis test must be performed. For using ANOVA, data should follow normality, so normality should be tested. Later on, pair-wise tests can be applied between groups, and p value will be corrected.*

Reply: Thank you for this important point and suggestion. As suggested by the reviewer, we re-analyzed the statistics and performed comparative analysis. Statistical comparisons included ANOVA for >2 groups (after testing data for normality) and subsequent pair-wise tests where applicable. Importantly, this re-analysis of the data did not change the level of statistical significance of our findings.

15. *Please check the reference, for the main manuscript only 5 were used, while for M&M 23. If it necessary, include more as the consideration of the authors.*

Reply: We revised the references for the main manuscript as suggested by the reviewers, which now has 71 references.

16. *Why experiments from Figure 4P-Q were done only for glutamine and not for butyrate?*

Reply: We used the cytotoxicity assay as a final approach to explore the mechanisms by which glutamine prevented the epithelial injury of neonatal mice independently of IL-10. This approach was chosen because the in vitro studies shown in Fig. 2F and G did not point to a direct effect of butyrate on immune cells.

Minor comments:

17. *Line 77, please add the reference of the previous publication.*

Reply: We have added the reference (Reference #6: PLoS One 2017;12: e0182089).

18. Line 88, PBS is control group or water? Please unify to make it clear.

Reply: We have revised the manuscript to consistently report that PBS was used in the control group for all intraperitoneal injections and oral treatments of newborn mice.

19. Line 94, “extrahepatic bile ducts of neonates from butyrate-fed mothers were patent”, please check, the sentence looks odd.

Reply: We have changed the above sentence to state that “...these mice had patent extrahepatic bile ducts and only mild inflammation despite RRV infection...” (Page 5).

20. Line 123-125, “rotavirus-infected mice that did not develop biliary obstruction” please include the name of the groups such as “butyrate-RRV resistant” for example. So that the groups in the figure match with the sentence in the text.

Reply: We revised the sentence, which now reads “...the subgroup of rotavirus-infected mice that did not develop biliary obstruction (“resistant” subgroup) was similar to their butyrate-fed mothers and to rotavirus-naïve mice, and differed from the subgroup of mice that developed the disease (“diseased subgroup”; Fig. 3D,E)....” (Page 7).

21. Line 127, delete female, pregnant are always female.

Reply: Deleted.

22. Line 128-129, “predominant enrichment of Firmicutes and Bacteroidetes with a decrease in Proteobacteria”, please review this sentence, proteobacteria seems to be decrease when diseased and resistant are compared.

Reply: We revised the sentence as suggested on Page 7. It now reads “...predominant enrichment of Firmicutes and Bacteroidetes with a decrease in Proteobacteria in mice resistant to RRV-induced biliary injury when compared to those with the diseased phenotype (Fig. 3G,H)....”

23. Line 127, what about the unique 1315 OTU's unique from the pregnant mice? These are almost 70% of the total OTU's from the mothers, and mothers and disease-resistant babies only share 30% of their mothers.

Reply: We agree that it may be valuable to examine the 1315 OTUs that are unique to pregnant mice as they may be of value to the mother. Such an analysis, however, is outside of the goal of our analysis which seeks to study which component of the mother's microbiome is reflected (or transmitted) to the offspring. To facilitate a view of the mother's unique OTUs, we added an initial analysis in Supplementary Table S5 of the revised manuscript and is mentioned on Page 7: “...Correspondingly, analysis of the 1315 OTUs detected exclusively in fecal specimens from butyrate-fed pregnant mice showed the largest abundances for Bacteroidetes and Firmicutes (Table S5)....”

24. Please be consist through the text, as use IL or Il for interleukin, but not both.

Reply: We have carefully edited the manuscript to make sure that we consistently apply the conventional use the capital letters “IL” to denote the protein and small “Il” when we refer to the gene name/mRNA expression.

25. Please change rpm units by xg, otherwise your experiments cannot be reproducible for other research groups.

Reply: We have followed the reviewer's suggestion and express the centrifuge rotational units as "g".

26. *Line 252, you have an underscore after a dot.*

Reply: We have corrected this error. Thank you.

27. *Please define mCL.*

Reply: mCL is the name given to a murine cholangiocyte cell line. We revised the manuscript to properly define "mCL" (Page 18).

28. *Figures footnotes, for mean +/- SD, it does not look to be SD, but to be SEM, please check.*

Reply: We have double-checked all the data and confirmed they are \pm SD and not SEM.

29. *Lines 478 and 480, please add the reference number of the data submitted to the electronic libraries.*

Reply: We have added the reference number of the submitted data:

1) MTBLS2171: <https://www.ebi.ac.uk/metabolights/studies>,
2) <https://www.ebi.ac.uk/ena/submit/sra/#studies>. Primary accession PRJEB40649.
Secondary accession: ERP124306.

30. *For all figures, in the immunohistochemical pictures, please add the scale.*

Reply: We have added the scale bars for all images as suggested.

31. *Figure S1. Line 26 "show increased TREG and IL-10+TREG cells", these type of line are results, they should not be included as part of the figure.*

Reply: We have removed the sentence from the figure legend. Thank you for the suggestion.

REPLY TO REVIEWER #2

1. *The authors show that feeding mouse dams butyrate or feeding the mouse pups butyrate protects the pups from RRV-induced BA. It is quite possible that the reason for this is a protection against RRV infecting cholangiocytes or reducing the duration of this infection. This was not evaluated. It would be important for the authors to perform plaque assays following RRV infection to determine if the frequency of active infection and the duration of infection was similar or different in the mice from butyrate fed dams vs. controls.*

Reply: Following the insightful suggestion, we performed new experiments and quantified RRV titers by focus-forming assays. As shown in the new Panel G of Fig. 1, the abundance of RRV in livers and extrahepatic bile ducts (EHBDs) is similar in newborn mice from water- or butyrate-fed mothers, which suggested that butyrate did not protect tissues from RRV infection or modify tissue titers. Similarly, we performed new experiments following butyrate administration to RRV-infected newborn mice and found no change in RRV titers (new Panel D of Fig. 2). These new results are discussed on Pages 5 and 6 and the experiments are described in the updated Methods section (pages 15–16).

2. *Did all of the mice in each of the litters from each of the butyrate fed dams respond similarly to protection from BA? This would be expected since each fetal mouse should have been exposed to the same intrauterine environment as well as maternal fecal microbiome after birth.*

Reply: Not all neonatal mice from butyrate fed dams responded similarly to protection from BA. We documented the reproducibility of this finding in 10 separate experiments: 10 different dams, which generated 10 litters with 5-9 pups per litter (average of 7.2 pups per litter). The most common finding within each litter was the presence of a subgroup that was resistant to BA and the other subgroup displaying the BA phenotype – this is now more clearly depicted in the new Fig. 1 which includes a graphic representation of the experimental design and findings (Fig. 1A). Our current interpretation is that the microbiome changes are an important factor in the pathogenesis of biliary atresia, but the development of the microbiome may not be uniform among all pups of the same litter, perhaps with variable levels of enrichment within each pup.

3. *It is assumed that the intrahepatic cell isolations in the mouse pups was from pooled livers. The authors should specify the pooling strategy that was used.*

Reply: The cell isolations were from individual mouse livers. This was important because we wanted to control for the “resistant” or “diseased” subgroups within each litter and because we wanted to estimate the numbers of cells per liver. We revised the Methods section to more clearly describe our experimental approach (Pages 16 and 17).

4. *How was butyrate administered “orally” to RRV infected mice on day 2 of life and thereafter?*

Reply: Oral administration of butyrate to 2-day old newborn mice was performed using a modified version of the intra-esophageal gavage protocol reported previously by *Francis et al. J Vis Exp; 2019* (Reference #8). In brief, we used a 25µl Luer tip Hamilton syringe and a 26-gauge, 1.25 mm ball diameter flexible catheter. The procedure involved holding the newborn between the thumb and index finger and inserting the needle at a 45° angle, which is allowed to slide naturally downward the esophagus, at which time the butyrate solution is administered. Following gavage, the needle is gently withdrawn, and the pup is observed until normal activity and breathing patterns are regained. No spillover or loss of butyrate solution was observed. This procedure was practiced extensively before applying to the experiments performed for this manuscript – now described on the Materials Section, Page 15.

5. *The decreased IL-10 from hepatic mononuclear cells from RRV infected pups that were incubated with butyrate is not consistent with the statement in the abstract that butyrate added to lymphocytes suppresses their activity. This should be corrected in the abstract.*

Reply: The reviewer is correct. In response, we have revised the abstract to accurately reflect our experimental results. It now reads: “...In vitro, butyrate did not induce IL-10 expression by activated hepatic mononuclear cells, while glutamine blocked cholangiocyte lysis by activated hepatic natural killer cells...”

6. *A relatively small number of animals had liver biochemistries measured compared to the total number in Figure 1A. Were liver biochemistries performed on random mice*

from separate litters to make the results (n=5-7) more representative of the much larger cohort (n=58-72) that were studied?

Reply: Yes, liver biochemistries were measured on plasma samples from mice selected randomly from separate litters to make sure the results were more representative of the much larger cohort. This is described now in the Methods on Page 16 as: "...mice were selected randomly from separate litters to make sure the results were more representative of the much larger cohort..."

7. *How do the authors explain that the hypoxanthine in stool appeared to be protective in the BA mice but was increased in stool of BA human infants?*

Reply: Hypoxanthine (like glutamate/glutamine) was increased in fecal supernatants of pups born to butyrate-fed dams (above the levels in water-fed dams). These pups had not been exposed to RRV; thus, we had no evidence it was protective. The finding did raise the possibility that the enrichment of the intestinal content with hypoxanthine (and glutamate/glutamine) would protect pups from BA. Testing this possibility, however, we found no evidence that hypoxanthine was protective in RRV-infected pups.

8. *In several figures, liver and bile duct histology from one animal in each group is shown. How representative was this? How many animals had liver histology and bile duct histology measured for each of these Figures, and what percent showed similar reductions in inflammation, obstruction, etc.? How did the authors quantitate changes in liver histology (scoring system)?*

Reply: The histology figures are representative of each experimental group, with several livers and bile ducts showing consistent features within each group. As for the quantification of changes in the liver histology, we used the scoring system published by us previously (Reference #47 and #48) and now include detailed histological quantification of liver and EHBD sections in Supplemental Figure S1.

9. *The cholangiocyte protective effect of glutamine when incubated with NK cells was quite interesting. Since glutamine is one of the precursor amino acids of glutathione, did the authors measure glutathione levels in the cholangiocytes following glutamine incubation? Did glutathione levels increase in the liver of glutamine supplemented rat pups vs. the controls or the inosine supplemented pups? There are previous studies suggesting that glutathione and its pathways may be low in liver of human infants with BA, so this should be examined in these experiments and discussed in the Discussion... It would be interest to use N-acetyl cysteine in these experiments as another means of increasing glutathione levels to determine if this would protect against NK-induced cholangiocyte lysis.*

Reply: We performed new experiments as suggested by the reviewer. First, we measured glutathione levels in cholangiocytes cultured with glutamine and found higher levels of GSH with increasing concentrations of glutamine (new Fig. 7G). Second, we obtained NK cells from livers of RRV-challenged neonatal mice and co-cultured these cells with a murine cholangiocyte cell line (mCL) in the presence or absence of n-acetylcysteine (NAC). We found that preincubation of cholangiocytes with NAC almost completely abrogated cholangiocyte lysis. These results correlated with the ability of glutamine to suppress NK cell mediated cytotoxicity (Fig. 7H). And third, we inoculated RRV into new groups of neonatal mice and assigned them to glutamine or saline

treatment using the previous protocol. We then measured GSH in liver and EHBD tissues. While RRV infection significantly decreased the concentration of GSH in the liver and EHBD, the administration of glutamine increased GSH levels in both tissues and approached the levels of uninfected controls (Fig. 7I,J). These results are described on pages 10 and 11 of the revised manuscript.

REPLY TO REVIEWER #3

Major comments

1. *An important concern is the physiological relevance and impact of orally administering high butyrate concentrations. The authors are suggesting the conclusions from this work resemble the case of bacterially produced butyrate in the colon (see lines 217-218, 226-228)... What is the justification of using 200 mM butyrate? ...What is the concentration of butyrate in portal vein?*

Reply: The reviewer makes excellent points. We agree the butyrate is not part of human diet and that it is produced in the colon. Our selection of butyrate and the administration protocol is based on two premises. First, we use butyrate as a tool for proof-of-principle experiments to explore the potential role of how changes in the intestinal microbiota may influence the pathogenesis of a neonatal-onset liver/bile duct injury in biliary atresia. The reviewer is right that oral administration would likely affect the intestinal microbiota, which was our intent. Second, the selection of butyrate is based on our earlier study reporting an increase in butyrate-producing bacteria in neonatal offspring of dams treated with antibiotics (Ref #6). In selecting the dose, we used doses from previous reports by other investigators (Reference #40, #41 and #42). In regard to the use in humans, doses of 100 mM butyrate or 200 mM SCFA (which included butyrate) administered as rectal enemas were used in previous studies (Reference #43 and #44). Although the concentration of butyrate in portal vein in humans is unknown, studies in rodents reported concentrations of $237 \pm 72 \mu\text{mol}$ (Reference #45) and $110\text{--}325 \mu\text{M}$ (Reference #46), depending on the experimental approach. Despite the rationale and these studies, we see the value of the reviewer's and revised the manuscript to cite the references in the Methods section.

2. *The presentation of the microbiome analyses is not sufficiently clear to warrant the simple conclusion that maternal butyrate-confers maternal producing microbiome in neonates... Can the authors present any evidence that these changes are the cause of the disease rather than a manifestation of it? If not, then the text needs to be nuanced and toned down to discuss both scenarios.... Maybe also the microbial profiles of the diseases vs resistant should be shown in addition to the subsets shared by different as shown in Fig. 2F (so just a comparison of the profiles of all identified OTU). ...Orthogonal transfer of microbiota is well established, and indeed the effects observed in mice, could well be microbiota mediated. However, can the authors preclude that butyrate or other metabolites that confer resistance are transferred to the offspring via mother's milk?*

Reply: Direct evidence would require fecal transplantation experiments in neonates, which in itself has potential for major flaws because of the mechanical challenges in trying to administer fecal matter to very small neonatal mice (spilling of fecal material, potential trauma to mouth, etc) as well as the dynamic differences in the development of the microbiota in the early postnatal period. In our experimental design, we overcame these

challenges by asking the question of whether metabolites in the fecal supernatant could protect neonatal mice from biliary injury. We accept the comments of the Reviewer to tone down the interpretation (second paragraph of the Discussion).

In regard to presenting the microbial profiles of the “disease” vs “resistant” subgroups of mice, we performed this comparative analysis and include the taxonomic groups between pregnant, resistant and diseased mice in Supplementary Table S3. All the unique OTUs identified from the diseased and resistant groups of mice are now presented as Supplementary Table S4 and mentioned in the Results section (page 7 of the revised manuscript).

The reviewer is correct that butyrate or other metabolites may be transferred to the offspring via mother’s milk, a possibility that we did not address directly. However, our experimental approach to administer butyrate to neonatal mice demonstrate that butyrate is able to directly protect neonatal mice from RRV-induced biliary atresia. We have discussed the reviewer’s important points as we did not directly quantify butyrate in mother’s milk or other tissues (pages 13 and 14).

3. *The choice to administer Gln rather than Glu and Inosine rather than hypoxanthine is puzzling... Therefore, this choice needs to be justified. Also, I am not sure what the performed pathway analysis is adding as it has been presented... The authors should attempt to see if the “enriched pathways” they describe are somehow associated with the taxonomic groups identified as signatures for resistance.*

Reply: *In regard to glutamine rather than glutamate*, both metabolites are increased in the fecal supernatant of mice born to butyrate-fed mothers (Fig. 4G). Our rationale for choosing glutamine is based on a previous report of low glutamine levels in plasma of children with biliary atresia (ref: Zhao D, et al. PLoS One 2014;e85694) and on three additional features of glutamine: 1) its role in influencing intracellular bio-energy utilization, whereas glutamate (with its negative charge) has specific biological functions such as in neurotransmission; 2) the described role of glutamine in modulating Treg cell function and ox-redox balance via GSH (two important mechanisms of pathogenesis in biliary atresia); and 3) the safety profile of glutamine with a long-history of use in parenteral nutrition in humans. *As for the use of inosine*, it too is increased along with hypoxanthine, but at lower concentration (Fig. 4G). Our selection of inosine is because: 1) hypoxanthine is poorly soluble in water, cannot be administered parenterally, and is not intended for clinical use due to its potential toxicity; 2) inosine serves as a precursor of hypoxanthine; and 3) inosine has immunoregulatory properties (biological processes implicated in pathogenesis of biliary atresia). We have revised the manuscript to include this rationale (pages 9–10).

As suggested by the reviewer, we applied the same analytical methods used to generate the “enriched pathways” in the human microbiome data to the taxonomic groups identified as signatures for resistance. We found that the pathways with highest levels of enrichment included L-glutamate and L-glutamine biosynthesis, adenosine nucleotide degradation, and purine nucleotide degradation (Supplementary Fig. 4).

4. *The Urea cycle in some G-positive Bacilliales has been proposed to be activated in response to acidic stress, (see 10.1128/JB.02517-14), which maybe be interesting in the context of pH changes in the small intestine related to biliary injury? Also, the initial*

hypothesis on butyrate, maybe should be evaluated by looking at the enrichment of butyrate synthesis genes in the cohort.

Reply: Thank you for the comment on how pH changes in the small intestine may relate to biliary atresia. This is an interesting possibility that is outside the scope of the current work and will be best addressed in future experiments. As for the enrichment of butyrate synthesis genes, we followed the reviewer's recommendation and performed the suggested analysis using the metagenomic data for infants with biliary atresia and controls. We found significantly decreased levels of the enzymes *Butyrate kinase* (-3.2-fold, $P=0.0034$) and *Gamma-glutamyl-gamma-aminobutyrate hydrolase* (-1.6-fold, $P=0.033$) with marginally decreased *Acetate CoA-transferase* (-0.12-fold, $P=0.098$) and *4-hydroxy-2-oxoglutarate aldolase* (-1.6-fold, $P=0.083$) in biliary atresia. The decreased expression levels in fecal specimens point to altered butyrate synthesis pathways and may suggest a potential benefit for butyrate supplementation. **This analysis is not included in the manuscript, but we will be glad to add if this reviewer finds it necessary.**

Analytical methods: Aggregation of the relative abundances of UniRef90 gene families to Enzyme Commission (EC) number reactions (level 4) using the HUMAnN2 `humann2_regroup_table` command and default database, followed by the analysis of differentially abundant EC reactions using the ALDEX2 package v1.16.0. Enzymes involved in Butyrate reactions seen in >20% of samples were retained for testing and converted to counts per million (cpm). Tests for differential abundances were conducted using the Wilcoxon rank-sum test averaged over 128 Monte-Carlo instances drawn from the Dirichlet distribution. Reactions with an FDR corrected p-value of <0.05 were considered differentially abundant.

Minor:

5. P7, L144 the "...greatest differential expression" is not appropriate, this is abundance and not expression you are measuring.

Reply: Thank you for identifying the inaccuracy. We have changed the text appropriately.

6. Line 252, there is an issue with a funny dash.

Reply: The dash has now been removed from the updated manuscript.

REVIEWER COMMENTS

Reviewer #1 (Remarks to the Author):

To the authors:

I would like to congratulate the authors, they have addressed all the comments successfully.

General comments: The manuscript is very well written and it is from very high interest for the scientific community. I enjoyed the lecture and the findings, and I consider this paper is of high interest and contributes greatly to the understanding of microbiota role by maternal regulation in biliary disease in neonates.

However, I detect very few minor details:

1. Line 55. *In vitro* must be in italics.
2. Abstract. I would suggest to reduce a little the introduction to the topic, so that, authors can include a key finding from the human neonates analysis.
3. Line 106. Figure S1A has not been mentioned in the manuscript, please add.
4. Lines 140-141. Please check this sentence, I do not fully understand what you mean. You mean that resistant and pregnant butyrate-treated mice share higher number of OTUs compared to diseased mice group?
5. Lines 141-142. Please follow a numerical order for Tables. I strongly recommend to re-arrange the tables, and start with Table S1 and S2, instead of S3 and S4.
6. Line 157. After resonance please add "spectroscopy"
7. Line 177. Figure S5A has not been referred in the text please add it.
8. Lines 178-179. "in younger infants" means compared to the age of the neonate mice? please complete the sentence.
9. Line 232. For "the previous protocol", please add a reference or a Figure from the paper.
10. For table S4, please add the total numbers of taxonomic groups for resistant and diseased mice groups.

Reviewer #3 (Remarks to the Author):

The revised manuscript is improved. However, the microbiota-butyrate link, remains shaky. Despite the toned down microbiota angle, the reader still gets an impression of a mechanistic link between bacterial-butyrate production and the resistance to disease in the revised manuscript, which is not warranted from the data.

L253-259 in the discussion exemplify this issue, which I do not think is supported due to the following reasons:

1. The most obvious part of the data in mice is the decrease in Proteobacteria, which are inhabitants of the small intestine. Similarly, in the human cohort, there is a decrease in small intestinal residents from both Proteobacteria and Firmicutes (from the class Bacilli). The author's analysis in the revised manuscript shown only marginal changes in the butyrate encoding genes. Therefore, the data do not show a clear robust link to specific increase in butyrate producers, but a general higher relative abundance of colonic bacteria as opposed to small intestinal residents.
2. The authors control group are mice fed with water and not with another acid, e.g. acetate or citrate. Therefore the changes in microbiota, could be merely due to unspecific acidification of the small intestine, whereas the colonic bacteria that are more tolerant to acid are less affected. A similar pattern is also observed in humans (could be due to other reasons) that colonic bacteria increase and small intestinal Proteobacteria and Firmicutes are decreased in healthy individuals.
3. Lines 259-261 in the discussion "Of note, approximately 85% of the total butyrate-producing capacity is represented by members of the Bacteroidetes, Lachnospiraceae, and Ruminococcaceae²⁶, while Firmicutes constitutes a major source of butyrate in the colon²⁷" is not consistent as Lachnospiraceae and Ruminococcaceae ARE colonic Firmicutes, and indeed they are responsible for the majority for butyrate production (To a far less extent a few exotic Bacteroides). So this sentence needs to be rephrased to state what the cited reference are actually reporting.
4. Although the authors investigated the abundance of butyrate producing genes and added a list of OUTs for the different groups of mice, no link between the two things is made. The Phylum level reference are not sufficient, as especially in the humans, Firmicutes are also show decreased abundances.

In conclusion, if the authors want to pursue the investigation of the specific link between bacterial colonic production of butyrate and the disease, then they need to look at the correlation between the difference bacterial groups and the butyrate producing genes, e.g. show relative abundance based on genera (add a panel similar to Fig. 3G, but with genera instead. and show if the increase in Firmicutes and Bacteroides correlates to increase in butyrate production genes or if this is a general colonic bacteria increase? The same goes for the human cohort.

If this does not hold water this narrative should be changed to focus on what the data show, i.e. changes in microbiota based on biogeography, i.e. increase in colonic decrease in small intestinal bacteria (which in the mice case could be caused by unspecific acidification, although this cannot be verified as the right control is not performed).

The additional Table S3, is less useful as the abundances are not readily shown, maybe a heat map presentation would be more useful between the difference groups.

Reviewer #4 (Remarks to the Author):

I thank the authors for their reply and their hard work. I feel most of the comments have been addressed and I only have minor suggestions at this stage. Please note that any additional experiments are desired but not essential and left to editorial discretion at this late stage.

Comment 1: Minor point. For clarity and consistency with the rest of the figure please show in Fig.1G the Dis and Res subgroups as in 1D or 1E

Comment 2: I take the authors' point. Another explanation could be that BA is multifactorial and even the RRV infected mice show some level of heterogeneity with 20% surviving. Please elaborate on these possibilities in the manuscript (e.g. discussion)

Comments 3-7: Addressed fully. Many thanks

Comment 8: I would like slightly more information. Please provide in the figure legend number of mice, number of sections per mouse and number of fields per section assessed.

For each of the 9 subpanels of SFig. 1B-1C please provide a representative image to supplement the main figure (where applicable).

Comment 9: The concept of GSH depletion in BA and other cholestatic disorders is not new (e.g Wells group) and this could result in reduced glutamine in the bile and subsequently in the stool. Furthermore, cholestasis could lead to changes in the microbiome (which then become a consequence rather than a cause for the disease). Adding a human control group of newborns with cholestatic liver disorders in Fig 5 and a mouse bile duct ligation control in Fig. 3 could help to ensure the changes observed are the cause rather than the result of the disease. These experiments are desirable but not essential at this stage. However, if they are not performed please discuss this limitation in the discussion.

RESPONSE TO REVIEWERS

Once again, the comments and questions from the reviewers have been very helpful and insightful. To address them, we carried out additional experiments and also performed new data analysis. The manuscript has been revised to incorporate these changes in the text and Figures. By addressing the reviewers' remaining comments and questions, we have further improved the quality of the manuscript, as discussed below in the point-by-point response. All changes in the manuscripts are shown in red to facilitate re-review of the manuscript.

Manuscript #

NCOMMS-20-36817

Manuscript Title

Maternal regulation of biliary disease in neonates via gut microbial metabolites

REPLY TO REVIEWER #1

1. *Line 55. In vitro must be in italics.*

Reply: We revised the abstract to incorporate other suggestions (below) and the word “*in vitro*” is no longer included. Thank you.

2. *Abstract. I would suggest to reduce a little the introduction to the topic, so that, authors can include a key finding from the human neonates analysis.*

Reply: We revised the Abstract to incorporate the reviewer's suggestion, while keeping word counts within limits established by the journal. Thank you.

3. *Line 106. Figure S1A has not been mentioned in the manuscript, please add.*

Reply: Done – Added on Page 6, Line 108. Thank you.

4. *Lines 140-141. Please check this sentence, I do not fully understand what you mean. You mean that resistant and pregnant butyrate-treated mice share higher number of OTUs compared to diseased mice group?*

Reply: Yes. The reviewer is correct. To improve clarity, we have changed the sentence on lines 147-150 to: “Similarly, the microbial OTUs from the butyrate group that did not develop biliary obstruction after RRV (“resistant” subgroup) were similar to their butyrate-fed mothers and to RRV-naïve newborn mice, but differed from the subgroup of mice that developed the disease.”

5. *Lines 141-142. Please follow a numerical order for Tables. I strongly recommend to re-arrange the tables, and start with Table S1 and S2, instead of S3 and S4.*

Reply: Thank you for noting the discrepancy. We have revised the sequence of the supplementary tables and numbered them based on the numerical order as suggested.

6. *Line 157. After resonance please add “spectroscopy”*

Reply: Done (Page 9, Line 167). Thank you.

7. Line 177. Fig. S5A has not been referred in the text please add it.

Reply: We have now referred this and another new figure in the text (with different numbers due to changes in the manuscript): Figure S4A-B (Page 9, Line 171) and Figure S7A (Page 10, Line 191). Thank you.

8. Lines 178-179. “in younger infants” means compared to the age of the neonate mice? please complete the sentence.

Reply: As this section describes our work in human infants, we now use the words “human infants” to better clarify and avoid ambiguity (Page 10, Line 188).

9. Line 232. For “the previous protocol”, please add a reference or a Figure from the paper.

Reply: We have revised the text and now include the reference to Figure 6A (Page 13, Line 251).

10. For table S4, please add the total numbers of taxonomic groups for resistant and diseased mice groups.

Reply: As suggested by the reviewer, we have updated this table (now labeled Table S1) by including the total numbers of taxonomic groups (Phyla, Classes, Orders, Families, Genera and Species). The data are shown in separate tabs/sheets of an Excel file to facilitate the analysis by future readers of the manuscript.

REPLY TO REVIEWER #3

1. “Although the authors investigated the abundance of butyrate producing genes and added a list of OTUs for the different groups of mice, no link between the two things is made. The Phylum level reference are not sufficient, as especially in the humans, Firmicutes are also show decreased abundances... If the authors want to pursue the investigation of the specific link between bacterial colonic production of butyrate and the disease, then they need to look at the correlation between the difference bacterial groups and the butyrate producing genes, e.g. show relative abundance based on genera (add a panel similar to Fig. 3G, but with genera instead and show if the increase in Firmicutes and Bacteroides correlates to increase in butyrate production genes or if this is a general colonic bacteria increase? The same goes for the human cohort.”

Reply: We generated new results from the murine and human microbiome data to appropriately respond to the reviewer’s additional comments.

In the murine-based analysis, we mined the microbiome data to assess the link between the “resistant” phenotype with butyrate/butanoate pathways (new Fig. S5), followed by a test of correlation with the bacterial taxa signatures depicted in the cladogram (Figure 3G – Spearman correlation coefficients and p-values using the *rstatix* module in the R package). As shown in the new Fig. S6, there is a statistically significant relationship (colored boxes) between the bacterial taxa and the phenotype, with white squares

depicting non-significant associations. Next, using gene annotations from the *MetaCyc* database, we identified signatures of genes linked to the pathways of Acetyl-CoA and Lysine that are related to butyrate synthesis (Table S3; see Excel file). These results point to an enrichment in *Firmicutes* and *Bacteroides* taxa in resistant mice, with prominent butyrate-related gene signatures within identified pathways in this phenotype. We have updated the Results section on Page 10, Lines 180-185.

For the analysis of the human data, we examined the abundance of butyrate-related genes for the seven bacterial populations that are enriched in normal control subjects (and decreased in the stools of children with biliary atresia) and found the overexpression of butyrate-related enzyme genes in five species (*Flavonifractor plautii*, *Hungatella hathewayi*, *Clostridium neonatale*, *Bacteroides dorei*, and *Bacteroides fragilis*). These results are shown in new Fig. S7H and the gene lists for individual bacteria are included in the Table S5.

These additional data point to an enrichment of butyrate-associated genes in the microbiome of newborn mice “resistant” to the experimental phenotype of biliary atresia and of normal human controls (without biliary atresia).

2. *The authors control group are mice fed with water and not with another acid, e.g. acetate or citrate. Therefore, the changes in microbiota, could be merely due to unspecific acidification of the small intestine, whereas the colonic bacteria that are more tolerant to acid are less affected. A similar pattern is also observed in humans (could be due to other reasons) that colonic bacteria increase and small intestinal Proteobacteria and Firmicutes are decreased in healthy individuals.*

Reply: The reviewer’s comments identified an important variable in our studies. To control for this variable and investigate the potential influence of an unspecified acidification of the intraluminal environment on the disease outcome, we performed new experiments feeding pregnant mice with propionic acid (sodium propionate, 200 mmol/L in drinking water, pH adjusted to 7.5) using the same experimental protocol for butyrate. Pregnant mice tolerated the feeding well. We found that propionic acid feeding during gestation did protect neonatal mice from RRV-induced biliary atresia, as evidenced by the uniform onset and progressive jaundice, 100% mortality by day 14, high serum ALT and bilirubin, and complete obstruction of extrahepatic bile ducts, and portal inflammation (new Fig. S2A-F). These experiments add further support for the more specific effects of butyrate in the protection of newborn mice to biliary injury. The results are included in the revised manuscript on Page 7, Lines 111-119 and in the Methods section on Page 17, Lines 341-344.

3. *Lines 259-261 in the discussion “Of note, approximately 85% of the total butyrate-producing capacity is represented by members of the Bacteroidetes, Lachnospiraceae, and Ruminococcaceae²⁶, while Firmicutes constitutes a major source of butyrate in the colon²⁷” is not consistent as Lachnospiraceae and Ruminococcaceae ARE colonic Firmicutes, and indeed they are responsible for the majority for butyrate production (To a far less extent a few exotic Bacteroides). So this sentence needs to be rephrased to state what the cited reference are actually reporting.*

Reply: Thank you for the comment. We have re-phrased the sentence according to the reviewer's suggestion as follows: "Of note, approximately 85% of the total butyrate-producing capacity in the colon is represented by *Firmicutes* (including *Lachnospiraceae*, and *Ruminococcaceae*) and to a lesser extent by the *Bacteroidetes* lineage" (Page 14, Lines 278-280).

4. *Although the authors investigated the abundance of butyrate producing genes and added a list of OTUs for the different groups of mice, no link between the two things is made. The Phylum level reference are not sufficient, as especially in the humans, Firmicutes are also show decreased abundances.*

Reply: This important comment about the association of butyrate-related genes, selected bacterial population, groups of mice with "diseased" and "resistant" phenotypes, and humans with biliary atresia and control was addressed in #1 above.

5. *The additional Table S3, is less useful as the abundances are not readily shown, maybe a heat map presentation would be more useful between the difference groups.*

Reply: In view of the large data and the new analysis summarized above, we agree with the reviewer that this data is less useful and has been deleted from the revised manuscript.

REPLY TO REVIEWER #4

1. *I thank the authors for their reply and their hard work. I feel most of the comments have been addressed and I only have minor suggestions at this stage. Please note that any additional experiments are desired but not essential and left to editorial discretion at this late stage.*

Minor point. For clarity and consistency with the rest of the figure please show in Fig.1G the Dis and Res subgroups as in 1D or 1E.

Reply: Thank you for the positive feedback. In regard to Fig. 1G, the liver and bile ducts used to quantify RRV were harvested from neonatal mice at day 7 after RRV infection (time of bile duct injury). At this time-point, the identification of subgroups of neonatal mice as "diseased" or "resistant" phenotypes is not possible, with the phenotypes only becoming apparent 10-14 after infection. As shown in the figure, the similar virus titers in neonates from water- and butyrate-fed mothers show that the protection against the disease is not related to the tissue viral load.

2. *I take the authors' point. Another explanation could be that BA is multifactorial and even the RRV infected mice show some level of heterogeneity with 20% surviving. Please elaborate on these possibilities in the manuscript (e.g. discussion).*

Reply: The reviewer is correct in that 20% survival represents the relatively small phenotypic heterogeneity in the neonatal mouse model. We have included this perspective in the first paragraph of the Results (Page 6, Lines 98-99).

8. *I would like slightly more information. Please provide in the figure legend number of mice, number of sections per mouse and number of fields per section assessed. For each of the 9 subpanels of SFig. 1B-1C please provide a representative image to supplement the main figure (where applicable).*

Reply: As suggested by the reviewer, we have updated the legends for Figures 1, 2 and 6 and the revised Fig. S1 to include the number of mice, number of sections per mouse and number of fields per section assessed. We have updated Panels 1B and 1C in Fig. S1 to include representative images of the extrahepatic bile ducts and livers for the appropriate experimental groups.

9. *The concept of GSH depletion in BA and other cholestatic disorders is not new (e.g Wells group) and this could result in reduced glutamine in the bile and subsequently in the stool. Furthermore, cholestasis could lead to changes in the microbiome (which then become a consequence rather than a cause for the disease). Adding a human control group of newborns with cholestatic liver disorders in Fig 5 and a mouse bile duct ligation control in Fig. 3 could help to ensure the changes observed are the cause rather than the result of the disease. These experiments are desirable but not essential at this stage. However, if they are not performed please discuss this limitation in the discussion.*

Reply: These are insightful comments, and we agree that the inclusion of another control group of diseased infants with other cholestatic disorders will allow for an evaluation of the impact of cholestasis in the microbiome. Such an experiment may be pursued by us or other investigators in the future. As for the mouse experiments, the inability to perform bile duct ligation in days 1-2 of life in extremely small neonatal mice limits the availability of this additional group. Performing the surgery in older mice will address primarily bile flow and microbiome outside the newborn period, which is outside the scope of our paper. We have revised the Discussion to include: "Our studies did not control for the influence of cholestasis in the microbiome composition of neonates and its impact on disease outcome. In humans, this can be done by the inclusion of a group of age-matched human infants with other types of cholestasis syndromes. Experimentally, this can be approached by bile duct ligation, but the inability to perform this type of surgery in 1-2 day old mice makes this model not particularly suitable." (Page 15, lines 307-312).

REVIEWERS' COMMENTS

Reviewer #3 (Remarks to the Author):

I appreciate and commend the hard work that the authors have put and the new analyses in the revised manuscript.

I have two important, but hopefully easy to address issues:

1. A key discrepancy that I have pointed out in my previous report has not been addressed. Namely, the dominance of clostridial Firmicutes on the levels of colonic butyrate as opposed to Bacteroides, whose contribution is marginal.

The authors in the rebuttal letter have addressed this (point 3 in the authors reply), but the words “and to a lesser extent the Bacteroidetes...” are missing in the article file, which still reads:

P14, L278: “Of note, approximately 85% of the total butyrate-producing capacity in the colon is represented by Firmicutes (including Lachnospiraceae, and Ruminococcaceae) and the Bacteroidetes lineage^{26, 27}”.

The above sentence should be corrected to include to “a lesser extent”, because the literature including the two cited articles is not supportive of an important role for Bacteroides in butyrate production. In fact, the Bacteroides species that are highlighted to be over-represented in the present study, i.e. Bacteroides fragilis, and dorei in humans and B. acidifaciens in mice are not known to generate any butyrate to my knowledge. The B. acidifaciens has been even reported to be associated with lower butyrate production in a previous study (J-Y Yang, et al. Mucosal Immunology 10:104–116, 2017). That is why in my previous report I have pointed out this and suggested that the Bacteroidetes are toned down with respect to butyrate production.

Perhaps this merits a comment from the authors in the paper to make a clear distinction between the well established butyrate production by Firmicutes and the tentative connection between butyrate production and specific Bacteroides species based only on genomic analyses.

2. I highly appreciate the control experiment with propionate that has been performed, but I just want the authors to confirm an important detail:

P7, L112: The revised manuscript mentions that the pH of the propionate has been adjusted to 7.5, whereas this is not mentioned for butyrate. Can the authors confirm whether or not the neutralisation has been done for both SCFAs or only for propionate?. If the neutralisation has been done for both, please mention that consistently in the results and the materials and methods. Please also mention what and how much was added to elevate the pH of water to 7.5.

Minor issue:

SI material P 13, L112-113: Legend of Fig. S7 reads “Individual bacterial species enriched in normal control 113 subjects overexpressing enzymes and genes of butyrate pathway (shown in Table S6).” It is not clear why overexpression of enzymes is mentioned? Is this not merely genomic data? I guess the authors mean relative gene abundances? Please correct or clarify this.

Reviewer #4 (Remarks to the Author):

I would like to thank the authors for their hard work. My comments have been addressed fully.

Manuscript #

NCOMMS-20-36817B

Manuscript Title

Maternal regulation of biliary disease in neonates via gut microbial metabolites

RESPONSE TO REVIEWER #3

1. *"P14, L278: "Of note, approximately 85% of the total butyrate-producing capacity in the colon is represented by Firmicutes (including Lachnospiraceae, and Ruminococcaceae) and the Bacteroidetes lineage26, 27".*

The above sentence should be corrected to include to "a lesser extent"...

Reply: We have revised the sentence as suggested.

2. *P7, L112: The revised manuscript mentions that the pH of the propionate has been adjusted to 7.5, whereas this is not mentioned for butyrate. Can the authors confirm whether or not the neutralisation has been done for both SCFAs or only for propionate?. If the neutralisation has been done for both, please mention that consistently in the results and the materials and methods. Please also mention what and how much was added to elevate the pH of water to 7.5.*

Reply: We revised the manuscript by stating we adjusted the pH for both SCFAs. The unadjusted pH of 200 mM sodium butyrate was 8.48 and that of 200 mM sodium propionate was 8.68. The adjusted the pH of both SCFA in drinking water to 7.5 using 0.1N Hydrochloric (HCl) acid. Results and Methods have been updated on pages 6 and 17.

Minor issue:

SI material P 13, L112-113: Legend of Fig. S7 reads "Individual bacterial species enriched in normal control 113 subjects overexpressing enzymes and genes of butyrate pathway (shown in Table S6)." It is not clear why overexpression of enzymes is mentioned? Is this not merely genomic data? I guess the authors mean relative gene abundances? Please correct or clarify this.

Reply: We agree with the reviewer's comment and have accordingly changed the legend in Supplementary Figure 7H as: "Relative abundances of bacterial species with butyrate producing capacity linked to enzymes and genes of the butyrate pathway (shown in Table S6)."